# Shifting Mineral and Redox Controls on Carbon Cycling in Seasonally Flooded Mineral Soils

Rachelle E. LaCroix[1], Malak M. Tfaily[2], Menli McCreight[1], Morris E. Jones[1], Lesley Spokas[1], and Marco Keiluweit[1,*]

[1] School of Earth & Sustainability and Stockbridge School of Agriculture, University of Massachusetts, Amherst, MA, United States
[2] Soil, Water and Environmental Science Department, University of Arizona, Tucson, AZ, USA

*Correspondence to*: Marco Keiluweit (keiluweit@umass.edu) and Rachelle E. LaCroix (rel268@cornell.edu)

**Abstract.** Although wetland soils represent a relatively small portion of the terrestrial landscape, they account for an estimated 20-30% of the global soil carbon (C) reservoir. C stored in wetland soils that experience seasonal flooding is likely the most vulnerable to increased severity and duration of droughts in response to climate change. Redox conditions, plant root dynamics, and the abundance of protective mineral phases are well-established controls on soil C persistence, but their relative influence in seasonally flooded mineral soils is largely unknown. To address this knowledge gap, we assessed the relative importance of environmental (temperature, soil moisture, and redox potential) and biogeochemical (mineral composition and root biomass) factors in controlling $CO_2$ efflux, C quantity and organic matter composition along replicated upland-to-lowland transitions in seasonally flooded mineral soils. Specifically, we contrasted mineral soils under temperature deciduous forests in lowland positions that undergo seasonal flooding with adjacent upland soils that do not, considering both surface (A) and subsurface (B/C) horizons. We found the lowland soils had lower total annual $CO_2$ efflux than the upland soils, with monthly $CO_2$ efflux in lowlands most strongly correlated with redox potential ($E_h$). Lower $CO_2$ efflux as compared to the uplands corresponded to greater C content and abundance of lignin-rich, higher-molecular weight, chemically-reduced organic compounds in the lowland surface soils (A-horizons). In contrast, subsurface soils in the lowland position ($C_g$-horizons) showed lower C content than the upland positions (C-horizons), coinciding with lower abundance of root biomass and oxalate-extractable Fe ($Fe_o$, a proxy for protective Fe phases). Our linear mixed effects model showed that $Fe_o$ served as the strongest measured predictor of C content in upland soils, yet $Fe_o$ had no predictive power in lowland soils. Instead, our model showed that $E_h$ and oxalate-extractable Al ($Al_o$, a proxy of protective Al phases) became significantly stronger predictors in the lowland soils. Combined, our results suggest that low redox potentials are the primary cause for C accumulation in seasonally flooded surface soils,

likely due to selective preservation of organic compounds under anaerobic conditions. In seasonally flooded subsurface soils, however, C accumulation is limited due to lower C inputs through root biomass and the removal of reactive Fe phases under reducing conditions. Our findings demonstrate that C accrual in seasonally flooded mineral soil is primarily due to low redox potential in the surface soil, and that the lack of protective metal phases leaves these C stocks highly vulnerable to climate change.

# 1 Introduction

Although wetland soils cover a relatively small portion of the Earth's land surface, they store an estimated 20-30% of the global soil C stocks (Mitsch et al., 2013). However, this C pool is under pressure from climate change, with increasing severity and frequency of droughts having substantial, yet largely unresolved consequences (Brooks et al., 2009, Fenner and Freeman, 2011). Increased droughts are expected to release previously stored C in wetlands back into the atmosphere (Gorham et al., 1991). Prior studies focused on C cycling in wetland soils have been primarily aimed at organic wetlands, such as peats and bogs (Laine et al., 1996) or coastal wetlands (Kirwan and Blum, 2011). Although freshwater mineral wetlands are estimated to contain 46 Pg C globally (Bridgham et al., 2006), they have received comparatively little attention.

Previous studies on C cycling in mineral wetland soils have focused on permanently flooded, rather than seasonally flooded sites (Krauss and Whitbeck, 2012). This is surprising given that seasonally flooded soils are characterized by greater microbial activity than permanently flooded wetlands, resulting in significantly greater greenhouse gas emissions (Kifner et al., 2018). Moreover, the consequences of climate change are expected to be most immediately evident in seasonal wetlands due to their dependence upon precipitation and seasonal groundwater recharge (Tiner, 2003). Seasonal wetlands can be considered as early warning ecosystems (Brooks, 2005); forecasting the impacts of climate change on permanently flooded wetlands. Thus, seasonally flooded wetlands represent essential endmembers to study the effects of climate change on permanently flooded wetland soils (Brooks, 2005).

Seasonal wetlands are geomorphic depressions in the landscape that have distinct hydrologic phases of flooding and draining (Brooks, 2005). These ephemeral wetlands are small (<1 hectare), but ubiquitous—comprising nearly 70% of all temperate forest wetlands in the US (Tiner, 2003). Seasonal flooding and drainage not only create biogeochemical "hotspots" for soil C and nutrient cycling along upland-to-lowland transitions, but also "hot moments" as these transition zones move seasonally (Cohen et al., 2016). These transition zones are also relatively large, as the generally small size of seasonal wetlands results in a disproportionally large and dynamic terrestrial-aquatic interface relative to total wetland area (Cohen et al., 2016). Determining the controls on C cycling within seasonally flooded mineral soils thus requires specific consideration of the fluxes and dynamics across these terrestrial-aquatic transitions.

Temperature and soil moisture are principal controls on C cycling in soils (Lloyd and Taylor, 1994; Wang et al., 2014). Temperature regulates biological and chemical reaction rates and thus regulates the rate at which decomposition of soil organic matter can occur (Davidson and Janssens, 2006). However, water saturation is a critical driver of organic matter (OM) decomposition processes in seasonally flooded systems (Neckles and Niell, 1994). Water saturation governs oxygen availability in soil pore spaces, as oxygen diffusion in water is 10,000-times slower than in air (Letey and Stolzy, 1964). The resulting oxygen limitations inhibit the activity of oxidative enzymes, such as phenol oxidase or peroxidase (Freeman et al., 2001; Keiluweit et al., 2016), which catalyze the depolymerization of higher-molecular weight OM compounds (e.g., plant-derived macromolecules) into smaller, assimilable compounds (Megonigal et al., 2003). Further, once oxygen is depleted, microbes rely on alternative terminal electron acceptors ($NO_3^-$, $Mn^{4+}$, $Fe^{3+}$, $SO_4^{2-}$) in heterotrophic respiration that yield less energy (Sutton-Grier et al., 2011). These thermodynamic constraints also dictate the types of organic substrate microbes are able to use in anaerobic heterotrophic respiration (LaRowe and Van Cappellen, 2011). Anaerobic conditions limit microbes to utilizing substrates that are chemically more oxidized, in turn preferentially preserving more chemically-reduced organic compounds (i.e., compounds with lower oxidation states) in soils and sediments (Boye et al., 2017; Keiluweit et al., 2017). While $CO_2$ emissions are often correlated with oxygen availability (or soil redox potential, $E_h$) (Koh et al., 2009), it is unclear to what extent such metabolic constraints result in the selective preservation of high-molecular weight, chemically-reduced OM in seasonally flooded systems where soils become aerated for prolonged periods.

Water saturation also impacts soil by controlling vegetation type and density—thus acting as an indirect control on root growth and activity belowground. Plant roots contribute to soil C stocks through rhizodeposition (exudates, secretions, dead border cells, and mucilage), dead root residues (Jones et al., 2009), and root-associated microbes (Bradford et al., 2013). Roots are the main contributors to C stocks in upland soils (Rasse et al., 2005), but the impacts of roots on soil C stocks in wetlands is less clear. Water saturation directly inhibits root growth due to the associated low dissolved oxygen concentrations (Day and Megonigal, 1993; Tokarz and Urban, 2015). Indirectly, water saturation in soil selects for plant species that can tolerate water stress—typically species that have developed advantageous traits to survive flooded conditions, such as shallow rooting systems (Tokarz and Urban, 2015). However, seasonally flooded soils select for an even smaller niche of plants, as

they must be tolerant of both upland and lowland conditions (Brooks, 2005). How root inputs from facultative upland-to-lowland plant species contribute to soil C content and chemistry in seasonally flooded soils is still not clear.

In addition to restricting microbial metabolism and root growth, water saturation influences the concentration and distribution of high surface area minerals that are potent sorbents for C in soils (Chen et al., 2017; Torn et al., 1997; Wagai and Mayer, 2007). In upland soils, iron (Fe) or aluminum (Al) (hydr)oxides protect OM from microbial decomposition, thereby contributing to C storage for centuries to millennia (Torn et al., 1997; Wagai and Mayer, 2007). In flooded soils, however, the rapid depletion of oxygen upon flooding can result in the reductive dissolution of Fe(III) oxides (Chen et al., 2017), potentially causing the mobilization of previously Fe-bound OM (Zhao et al., 2017). During water table drawdown, Fe(II) may be leached from the profile or re-oxidized to Fe(III) oxides upon re-oxygenation of the soil (Wang et al., 2017). While redox-mediated transformations of Fe(III) oxides and export of Fe(II) is a well-known phenomenon ("gleying") in seasonally flooded soils (Chen et al., 2017), their impact on mineral-associated OM has yet to be determined. Further, Al rather than Fe (hydr)oxides, are the predominate mineral phases contributing to OM retention in forested floodplain sediments because their solubility is controlled by pH rather than redox conditions (Borggaard et al., 1990; Darke and Walbridge, 2000), and may thus play a critical role in mineral protection in seasonally flooded soils (Chen et al., 2018; Chen and Thompson, 2018).

Water saturation thus likely governs C cycling in seasonally flooded soils through its combined impact on oxygen availability, root dynamics and mineral composition; but how the relative contribution of these biogeochemical controls vary across spatial and temporal gradients is still unknown. A recent study along hillslope transects in tropical forest soils representing an oxygen gradient (Hall and Silver, 2015), for example, found that a combination of Fe (II) (a proxy for reducing conditions), fine root biomass, and total Fe and Al concentrations explained the most variation of surface soil C content. How the relationships between C and important biogeochemical controls differ in systems that are subject to seasonal flooding is still in question, especially with depth (Barcellos et al., 2018).

In this study, we aimed to identify the predominant environmental and biogeochemical controls on $CO_2$ efflux, C content, and OM composition in seasonally flooded mineral soils. To accomplish this goal, we studied the impact of seasonal flooding on C cycling across complete soil profiles (0 to 1 m) in six replicated upland-to-lowland transects typical for the Northeastern US (Brooks, 2005). Our objectives were to (i) identify the environmental parameters that drive temporal

dynamics of $CO_2$ efflux in seasonally flooded soils and (ii) examine the relative importance of biogeochemical controls on C content and OM composition with depth. To accomplish our first objective, we related soil $CO_2$ efflux at three landscape positions (upland, transition, and lowland) spanning the transect over the course of a full drainage and flooding cycle to measurements of soil temperature, moisture, water table depth and redox potential. To accomplish our second objective, we

examined variations in C content and chemistry in both surface and subsurface horizons in relation to root distribution, mineral composition and redox potential. We hypothesized that seasonally reduced conditions upon flooding in lowland positions will result in lower $CO_2$ efflux, greater C accumulation, lower capacity of Fe/Al (hydr)oxides to protect OM, and the selective preservation of macromolecular or chemically-reduced OM compared to the upland position. We anticipated that the transition position would represent an intermediate between upland and lowland positions. Based on other reports for comparable sites

(Holgerson, 2015; Kifner et al., 2018), we expected methane production within these seasonal wetlands. However, $CO_2$ emissions were at least 15-times greater than methane production at those sites. While we acknowledge the disproportionate potency of methane as a climate-active greenhouse gas, this study aimed to determine the environmental and biogeochemical factors influencing C accrual or depletion in across the upland-lowland transition. We thus focused our monitoring efforts on quantitatively more important $CO_2$ emissions as the predominant C loss pathway.

**2 Methods**

**2.1 Site description**

Our study included six replicate forested wetlands in western Massachusetts that experience seasonal flooding through groundwater recharge; three sites are located at the UMass Experimental Farm Station in South Deerfield, MA, and three located within the Plum Brook Conservation area in South Amherst, MA. All sites consisted of soils that are glacially-derived

sandy loams classified as mesic Typic Dystrudepts. Vegetation is dominated by red maple (*Acer rubrum*) and white oak (*Quercus alba*) stands with understory vegetation primarily composed of cinnamon fern (*Osmunda cinnamonea*), Canada mayflower (*Maianthemum canadense*), reed canary grass (*Phalaris arundinacea*), and jewelweed (*Impatiens capensis*). Mean annual air temperature is 9°C and mean annual precipitation (rainfall and snowfall) is 120 cm (National Centers for Environmental Information (NCEI), National Oceanic and Atmospheric Administration (NOAA).

## 2.2 Field measurements

A transect in each seasonal wetland was delineated from an upland position to a lowland position (Fig. 1a-c). Three positions, termed "upland", "transition", and "lowland", along each transect were established as monitoring stations and for soil sample collection. Fig. 1 depicts the distance between the individual positions, which ranged from three to five meters. The replicate transects are within approximately 100 m of each other. The upland position does not experience flooding in a typical rainfall year, i.e., the water table does not rise above the soil surface. The transition position is located on the edge of the wetland, which typically does not get flooded in an average rainfall year but is under the influence of significant water table rise. The lowland position is in the lowest point of the transect and is flooded for several months throughout the year. Each landscape position was monitored for $CO_2$ emissions, soil temperature, volumetric moisture content (VMC) at 0 to 10 cm, water table depth, and $E_h$. Field measurements were collected weekly at each designated landscape position in all six seasonal wetlands from May through August, then monthly from September through April. A field portable automated gas flux analyzer (LI-8100A, LI-COR Biotechnology, Lincoln, NE) was used to measure rates of $CO_2$ emissions, on permanently installed PVC collars, soil temperature and VMC. Three measurements of $CO_2$ fluxes were taken at each individual PVC collar using observation times of one minute, with 15 second dead band and pre- and post- purge times. The standard deviation of three observations was calculated in the field and a 15 % threshold was used for acceptable measurements. If the resulting standard deviation of the three measurements was greater than 15 % subsequent measurements were taken until the threshold was met.

Water table fluctuations were monitored using slotted PVC pipes installed to depths of 50 cm, therefore water table depths below 50 cm were undetectable and not reported (Fig. 2b). Platinum-tipped $E_h$ probes were installed in triplicate at each depth of 15-, 30-, and 45-cm; each group (nine) of $E_h$ probes were accompanied with a single salt bridge filled with saturated KCl in 3% agar for the reference electrode. In total, each landscape position had 18 redox probes installed at each depth. $E_h$ was measured using a calomel electrode (Fisher Scientific, Pittsburg, PA) attached to a voltmeter and corrected to a standard hydrogen electrode by adding 244 mV to each reading (Fiedler et al., 2007).

**2.3 Soil sampling and analysis**

Soil samples were collected using hand augers in each of the six replicate wetlands along the defined transects that included the three landscape positions (upland, transition, and lowland). In each landscape position, we collected soil cores from designated horizons, resulting in six replicate cores for each horizon and position. Horizons in the upland position were classified as A (0-25 cm), B (25-55 cm), and C (55-84+ cm) horizons; in the transition position as A (0-28 cm), C (28-48 cm), and $C_g$ (48-69+); and in the lowland position as A (0-25 cm), C (25-35 cm), and $C_g$ (35-68+ cm) (Soil Survey Staff, 1999) (Fig. 1a). Coarse rocks and roots were removed from soil samples which were then sieved using standard 2 mm screens. Particle size distribution was determined using the pipette method outlined by Gee and Bauder (1986). Total C and N were determined with an elemental analyzer (Hedges and Stern, 1984). Extractable iron and aluminum concentrations were measured on each soil horizon from all three positions from the six pools (n=62) using ammonium-oxalate and citrate-bicarbonate-dithionite (CBD) extraction procedures (Loeppert and Inskeep, 1996). Ammonium-oxalate extractable Fe ($Fe_o$) and Al ($Al_o$) represent the poorly crystalline pool of Fe, while the CBD extractable Fe ($Fe_d$) and Al ($Al_d$) represent the total reducible Fe.

Root biomass was determined by taking soil cores in all six wetlands at each position along the designated moisture transects. The cores were taken at 0-20 cm, 20-40 cm, and >40 cm and resulted in six cores per landscape position horizon (e.g., upland A-horizons n = 6). Root biomass was determined using a USDA NRCS hand sieving method (Soil Survey Staff, 1999). The initial values of root biomass were used to estimate biomass values for each soil horizon using an equal-area quadratic spline equation (Malone et al., 2009; Spline Tool v2.0, ASRIS). Mean $E_h$ values for each soil horizon were also estimated using the equal-area quadratic spline equation (Malone et al., 2009).

To determine the relative abundance of specific C functional groups and degree of oxidation, soil samples were analyzed using C (1s) near edge X-ray absorption fine structure (NEXAFS) spectroscopy at the Canadian Light Source (CLS) in Saskatoon, Canada. Soil samples from individual horizons were gently ground, slurried in DI-$H_2O$ and pipetted onto clean In foils. After drying, C NEXAFS spectra were obtained using the spherical grating monochromator (SGM) beamline 11ID-1 (Regier, 2007). Step scan mode (0.25 eV steps from 270 to 320 eV) was used to minimize x-ray damage. A dwell time of 20 ms was used between scans. Individual spectra were collected at new locations on each sample for a total of 40 to 60 scans.

The beamline exit slit was set at 25 mm, and the fluorescence yield data was collected using a two-stage microchannel plate detector. The resulting spectra were averaged for each sample and the averaged spectrum was then baseline normalized to zero and then normalized the beamline photon flux ($I_o$) from a separate Au reference foil. Each spectrum was calibrated to the carboxylic acid peak (288.5 eV) of a citric acid standard. Pre-edge (270-278 eV) and post-edge (310-320 eV) and an $E_0$ (290

eV) values were used to perform an edge step normalization. Peak deconvolution was conducted in Athena (Demeter (version 0.9.25, 2006-2016); Ravel and Newville 2005) to determine the relative abundances of functional groups, with peak positions as described in Keiluweit et al. (2017). Gaussian peak positions, their full-width at half-maximum, and the arc tangent function were fixed. Peak height was set to vary freely during the fitting process. Parameters were adjusted until optimal fits for each spectrum were achieved and all spectra were fitted with these final parameters.

To determine the composition of bioavailable compounds that can potentially be used in microbial respiration (<600Da, Logue et al., 2016), water extracts of soil samples were collected on a 12 Tesla Bruker SolariX Fourier-transform ion cyclotron resonance mass spectrometer located at Environmental Molecular Sciences Laboratory (EMSL), a Department of Energy Biological and Environmental Research (DOE-BER) national user facility located in Richland, WA. Soil samples were extracted with ultrapure DI-$H_2O$ using one gram of soil and 10 mL of DI-$H_2O$ (1:10). The samples were sealed in 15 mL

conical tip tubes and shaken for one hour. Samples were then centrifuged and filtered using 0.2 μm syringe-filters. 100 μL of the filtrate was then added to 200 μL of methanol (1:2) and injected directly (in liquid phase) onto the instrument. A standard Bruker electrospray ionization (ESI) source was used to generate negatively charged molecular ions; samples were then introduced directly to the ESI source. The instrument was externally calibrated to a mass accuracy of <0.1 ppm weekly using a tuning solution from Agilent, which contains the following compounds: $C_2F_3O_2$, $C_6HF_9N_3O$, $C_{12}HF_{21}N_3O$, $C_{20}H_{18}F_{27}N_3O_8P_3$,

and $C_{26}H_{18}F_{39}N_3O_8P_3$ with an m/z ranging between 112 to 1333. The instrument settings were optimized by tuning on a Suwannee River Fulvic Acid (SRFA)) standard. Blanks (HPLC grade MeOH) were also ran at the beginning and the end of the day to monitor potential carry over from one sample to another. The instrument was flushed between samples using a mixture of water and methanol. The ion accumulation time (IAT) was varied to account for differences in C concentration between samples and varied between 0.1 and 0.3 s. Ninety-six individual scans were averaged for each sample and internally

calibrated using OM homologous series separated by 14 Da (–$CH_2$ groups). The mass measurement accuracy was less than 1

ppm for singly charged ions across a broad m/z range (i.e. 200 <m/z <1200). To further reduce cumulative errors, all sample peak lists for the entire dataset were aligned to each other prior to formula assignment to eliminate possible mass shifts that would impact formula assignment. Putative chemical formulas were assigned using Formularity software (Tolić et al., 2017). Chemical formulas were assigned based on the following criteria: S/N >7, and mass measurement error <1 ppm, taking into consideration the presence of C, H, O, N, S and P and excluding other elements. Peaks with large mass ratios (m/z values >500 Da) often have multiple possible candidate formulas. These peaks were assigned formulas through propagation of $CH_2$, O, and $H_2$ homologous series. Additionally, to ensure consistent choice of molecular formula when multiple formula candidates are found the following rules were implemented: we consistently chose the formula with the lowest error with the lowest number of heteroatoms and the assignment of one phosphorus atom requires the presence of at least four oxygen atoms. Peaks that were present in the blanks were subtracted from the sample data sets. Additionally, all single peaks i.e. peaks that are present in only one sample were removed and are not included in the downstream analysis. To further identify only "unique" peaks, we compared samples with the same group against each other to keep the peaks in the sample set that occur at least half of the samples for that group; peaks that occurred in less than half the samples were discarded from the final data set.

To visualize differences in SOM composition, compounds were plotted on a Van Krevelen diagram corresponding to their H/C (hydrogen to carbon) vs. O/C (oxygen to carbon) ratios (Kim et al., 2003). Van Krevelen diagrams provide a way to visualize and compare the average properties of OM and assign compounds to the major biochemical classes (i.e., lipid-, protein-, lignin-, carbohydrate-, - and condensed aromatic-like) (Kim et al., 2003). To identify the degree of oxidation of the SOM we calculated the nominal oxidation state of carbon (NOSC) (Keiluweit et al., 2017):

$$NOSC = -\left(\frac{-Z+4C+H-3N-2O+5P-2S}{C}\right) + 4 \tag{1}$$

in which C, H, N, O, P, and S correspond to stoichiometry values measured by FT-ICR-MS, and Z is equal to the net charge of the organic compound (assumed to be zero). We utilized the calculated double bond equivalent (DBE) to determine the degree of saturation of the identified C compounds, using the equation set forth by Koch and Dittmar (2006):

$$DBE = 1 + \frac{1}{2}(2C - H + N + P) \tag{2}$$

where C, H, N, O, P, and S correspond to stoichiometry values also measured by FT-ICR-MS. The DBE is a useful equation to determine the degree of unsaturation of organic carbon containing molecules, where higher DBE values indicates less H

atoms and a greater density of C-C double bonds. We also analyzed aromaticity of water extractable organic matter using a modified aromaticity index (AImod) to determine the density of C-C double-bonds, using the amended equation by Koch and Dittmar (2016):

$$AI_{mod} = \frac{1+C-\frac{1}{2}O-S-\frac{1}{2}(N+P+H)}{C-\frac{1}{2}O-N-S-P} \qquad (3)$$

which takes into consideration the contributions of heteroatoms and $\pi$-bonds. To identify shifts in average molecular weights we calculated molecular weight using stoichiometry values measured by FT-ICR-MS:

$$MW = (C \; x \; 12.011) + (H \; x \; 1.008) + (O \; x \; 15.999) + (S \; x \; 32.06) + (P \; x \; 30.974) + (N \; x \; 14.007) \qquad (4)$$

where each element is multiplied by its molar mass.

## 2.4 Statistical analyses

All statistical analyses and plots were done using Rstudio (Version 1.0.136, R Core Team 2015). The lm() function in Rstudio was used to perform linear regressions with the seasonal data to determine how various environmental parameters (soil moisture, water table depth and redox potential) predicted $CO_2$ emissions in the three landscape positions. Arrhenius models were used to determine how soil temperature predicted $CO_2$ emissions in the three landscape positions using OriginPro (OriginLab) with the equation (Sierra et al., 2012):

$k = Ae^{(-Ea/RT)}$       (5)

where Ea is the activation energy, A is the pre-exponential factor, R is the universal gas constant (8.314 J K$^{-1}$ mol$^{-1}$), and T is temperature in Kelvin (K). Relationships between total C and biogeochemical parameters were analyzed using linear mixed effects models with the lme4 package (Bates et al., 2015) in Rstudio. Regression analyses were conducted for the entire year-long dataset, and for the growing and non-growing seasons defined as May through September and October through March,

respectively. Two sets of mixed effects models were conducted; the first to identify which biogeochemical variables (root biomass, $Fe_o$, $Al_o$, clay, and $E_h$) predicted C content in the different landscape positions where wetland number (n=6) was a random effect and horizon (A, B/C, C/C$_g$) and one additional predictor variable were fixed effects. The second set of models aimed at identifying how the same variables predicted soil C at different soil depths, where wetland number was chosen as a random effect and landscape position (upland, transition, lowland) and one additional predictor variable as fixed effects. The

mixed effects models were performed individually with one fixed effect parameter in addition to the blocking factor of either horizon or landscape position. To correct for multiple testing effects, we used the Bonferroni correction factor where $\alpha_{corrected}$ is equal to 0.01. To test our hypotheses, differences among landscape positions were assessed individually for each set of horizons using analyses of variance (ANOVA) conducted in Rstudio (version 5.3.1) combined with Tukey's honesty significance difference (HSD) tests using R packages agricolae (de Mendiburu, 2017) and multcompView (Graves et al., 2015). Specifically, we compared values within surface (A), intermediate (B/C), and subsurface ($C/C_g$) horizons across the upland-to-lowland transect (Table3). Alpha values of 0.05 were used for letter designations indicating significant differences among the landscape positions. Statistical analyses were conducted on square-root transformed data when assumptions of normal distribution were not met.

## 3 Results

### 3.1 Seasonal dynamics

*Soil respiration.* $CO_2$ fluxes in each landscape position began to rise in May and peaked in September. Thereafter, $CO_2$ efflux in all positions gradually declined to a baseline level until November, and remained low through April (Fig 2a). Cumulative $CO_2$ emissions during the growing season substantially decreased across the upland-to-lowland transect (Table 1). The flooded period (February through June) of the lowland position extended into the first two months of the growing season. Relative to the lowland position (24 mol $CO_2$ m$^{-2}$ year$^{-1}$), cumulative $CO_2$ emissions were 38% greater in the transition position (33 mol $CO_2$ m$^{-2}$ year$^{-1}$), and 58% greater in the upland position (38 mol $CO_2$ m$^{-2}$). This general difference became even more pronounced when cumulative $CO_2$ emissions were normalized to C content (Table 1), with the upland position showing significantly greater emissions than in the transition (p-value <0.001; Tukey's HSD) and lowland (p-value <0.001, Tukey's HSD) positions. In the non-growing season, the transition position registered the largest cumulative $CO_2$ emissions (20 mol $CO_2$ m$^{-2}$), but there were no noticeable differences between the upland and lowland positions (16 and 15 mol $CO_2$ m$^{-2}$, respectively) (Table 1).

*Moisture dynamics.* As typical in seasonal wetlands in the Northeastern US (Brooks, 2005), the water table in all three positions was highest from January to July and lowest from August through December (Fig. 2b). The lowland position had the

greatest fluctuations in water table depth; the water table rose above the ground surface from February through June and dropped below the ground surface from July through January (-2 to -42 cm) (Table S1). The water table in the transition and upland positions showed similar seasonal dynamics, but the water table was significantly lower in the lowland position throughout the year. VMC generally followed water table fluctuations, although with less seasonal variation (Fig. 2c). Soil

moisture was consistently the greatest in the lowland position; during the growing season lowland VMC was 20% greater than the upland position (p-value < 0.05), and 15% greater in the non-growing season (p-value < 0.05) (Table S1).

*Redox dynamics.* $E_h$ values typically mirrored the hydrologic conditions of each landscape position, with the lowest values generally occurring from May to July and the highest values between October and February (Fig. 2d). The lowland position had the largest seasonal amplitude, with values of less than 100 mV between May and July and above 500 mV from

October to December. $E_h$ in the transition position only fell to values between 200 to 300 mV between May and July, and recovered to values near 600 mV by October. The $E_h$ values at the upland position remained above 450 mV throughout the entire year at 15 cm depth, but reached 400 mV or lower at 30 and 45 cm depths from May to July.

In sum, the rise in seasonal water table in lowland positions during the growing season was accompanied by decreased $E_h$ and $CO_2$ fluxes compared to the upland positions. As the water table dropped during the growing season $E_h$ and $CO_2$ fluxes

increased markedly.

## 3.2 Control on $CO_2$ fluxes

To determine which of the above environmental parameters best predict soil respiration across the hydrological gradient, we conducted a series of regression analyses (Fig. 3a-d). Regression analyses were carried out for subsets of the data

representing the (i) full year, (ii) growing season or (iii) non-growing season (Table 2).

*Soil temperature.* The strength of the relationship between $CO_2$ flux and soil temperature, as expressed by how well the data can be described using the Arrhenius equation (Sierra, 2012), decreased along the upland-to-lowland transect (Fig. 3a). Soil temperature explained the most variance of $CO_2$ fluxes in the upland positions throughout the full year (r = 0.72, p < 0.001) and the growing season (r = 0.62, p < 0.001) (Table 2). Comparing the three landscape positions, soil temperature

explained the least variation of $CO_2$ fluxes in all cases in the lowland position, especially in the growing season ($r = 0.45$, $p < 0.001$).

*Soil moisture.* As the relationship between $CO_2$ flux and soil temperature became weaker, that between $CO_2$ flux and water table depth gradually became stronger along upland-to-lowland transitions. $CO_2$ flux and water table depth (Fig. 2b) were significantly negatively correlated in the lowland positions in the full, growing season and non-growing season time periods (Table 2). The strongest correlation between water table depth and $CO_2$ flux occurred in the lowland position during the growing season ($r = -0.55$, $p < 0.001$), where it had a stronger relationship with $CO_2$ flux than soil temperature. Similarly, VMC and $CO_2$ fluxes were negatively correlated in the lowland position ($r = -0.51$, $p < 0.001$), with VMC showing a stronger relationship with $CO_2$ flux than soil temperature during the growing season (Fig. 2c)

*Soil redox potential,* In keeping with a strong relationship between moisture and respiration at the transition and lowland positions, $E_h$ was also most significantly correlated with $CO_2$ at the transition and lowland positions (Fig. 2d). $E_h$ was a comparable predictor for $CO_2$ flux in both the lowland ($r = 0.40$, p-value $<0.001$) and transition ($r = 0.41$, p-value $<0.001$) positions during the growing season, but had no correlation with $CO_2$ flux in the upland position (Table 2). The strong correlations between $E_h$ and $CO_2$ emissions were primarily limited to the lowland position.

In sum, $CO_2$ emissions in the upland position were most strongly correlated to soil temperature, while water table depth and VMC correlated more strongly with $CO_2$ fluxes in the lowland position during the growing season.

**3.3 Relating carbon concentration to root biomass and mineral composition across upland-to-lowland transitions**

To identify how roots and mineral composition affected C distribution across the upland-to-lowland transect, we examined C content in relation to root biomass, texture, extractable Fe and Al and $E_h$ (Table 3). Along the upland-to-lowland transects, C content in the surface horizons increased (p-value $< 0.05$) whereas content in the subsurface horizons decreased along the transect (p-value $< 0.05$) (Table 3). C content in the lowland position surface horizons were four times greater than the upland position surface horizons (p-value $< 0.001$), respectively. In contrast, the subsurface soils in the upland positions had nearly double the C content than in the transition and lowland positions (p-value $< 0.05$) (Table 3). Root biomass significantly decreased from the upland to the lowland positions (p-value $< 0.05$) (Table 3), with a 10-fold decline in both surface and subsurface horizons. Silt and clay content increased from the upland to the lowland positions, particularly in the

subsurface soil (+33%, Table 3), although shifts in silt and clay contents were also not statistically significant. Both $Fe_o$ and $Al_o$ significantly decreased along the upland-to-lowland transects in the surface horizons (p-value < 0.05). However, in the subsurface horizons, $Fe_o$ showed a two-fold increase from the upland to the lowland positions (albeit not significantly), while $Al_o$ showed a four-fold decline (p-value < 0.001) (Table 3). $Fe_d$ and $Al_d$ followed the trends of $Fe_o$ and $Al_o$ (Table 3), thus we

further limit our discussion to $Fe_o$ and $Al_o$. Overall, in the surface horizons, we found C content increased along upland-to-lowland transects, which was accompanied by a decrease in root biomass and extractable $Fe_o$ and $Al_o$. However, in the subsurface horizons C content decreased along the same transect as root biomass decreased.

## 3.4 Linear mixed effects models between total carbon and biogeochemical parameters

To predict the relative influence of roots, mineral composition and $E_h$ on C content in each landscape position, we

performed linear mixed effects models using total C as a response variable and root biomass, clay, $Fe_o$, $Al_o$ and mean $E_h$ in the growing season as predictor variables with horizon as a blocking factor (Fig. 4a). The relative importance of the predictor variables changed across the upland-to-lowland transects. In the upland positions, $Fe_o$ was the strongest predictor with the largest F-value (17.31, p-value < 0.001), followed by root biomass (13.31, p-value < 0.01). In the transition and lowland positions, however, only root biomass and particularly $Al_o$ were significantly correlated with C (p-value < 0.01; Table 4). The

model results show that as the importance of redox-active $Fe_o$ as a predictor for soil C content decreased along upland-to-lowland transects, the importance of $Al_o$ increased.

To predict the influence of the biogeochemical variables on soil C content with soil depth, we performed linear mixed effects models on the different horizons, using landscape position as a blocking factor (Fig. 4b). In the A-horizon, $E_h$ had the highest F-value and strongest correlation to C (6.31, p-value < 0.05; Table S3). In the lowest horizons, $Al_o$ was the only

significant predictor variable in the models (F-value = 16.10, p-value < 0.01, Table S3). These results indicate that, among the tested biogeochemical variables, $E_h$, a proxy for oxygen availability, has a predominant influence on C content in the surface soils, while $Al_o$ has the strongest influence on C content at depth.

### 3.5 Carbon chemistry across upland-to-lowland transitions

To examine variations in C chemistry along upland-to-lowland transects, we analyzed solid-phase and water-extractable OM. C (1s) NEXAFS spectra showed a general increase in abundance of chemically-reduced, solid-phase C across the upland-to-lowland transect in the surface horizons, but an opposite trend in the subsurface horizons (Fig. 5a, Table S4). Aliphatic and carboxylic C relative abundances were significantly different amongst the three landscape positions ($p$-value < 0.05). The relative abundance of chemically-reduced aliphatic C increased from the upland to the lowland position in the surface horizons ($p$-value < 0.001); though not statistically significant, its contribution also decreased gradually along the same transect in the subsurface horizons (Fig. 5a, Table S4). Generally, chemically-reduced aromatic C followed the same trend as aliphatic C. Chemically more oxidized carboxylic C decreased in the surface horizons from the upland to lowland positions ($p$-value < 0.01), yet increased slightly in the subsurface horizons along the same transect. As a measure of the degree of oxidation, we calculated carboxylic-to-aromatic C ratios (Fig. 5b), with higher ratios indicating a greater degree of oxidation. Although the ratios were not significantly different among the landscape positions ($p$-value > 0.05), noteworthy trends were found. In the surface horizons, the ratio gradually decreased across the upland-to-lowland transects in the surface horizons (Fig. 5b, Table S4). In the subsurface horizons, the opposite trend was observed, and the ratio steadily increased from the upland C-horizons to the lowland $C_g$-horizons (Fig. 5b, Table S4).

To assess changes in oxidation state and molecular weight of compounds more readily available for microbial respiration, water extracts of all samples were analyzed by FT-ICR-MS (Fig. 6a-b, Fig. S1-3, Table S5, Table S6). The composition of water extractable OM was statistically indistinguishable across the transect ($p$-value > 0.05), but some general trends were noticeable. Both the modified aromaticity index ($AI_{mod}$) and the average molecular weight of the detected compounds showed gradual increases across the upland-to-lowland transitions in the surface horizons (Fig. 6a, Table S6). Paralleling that change, the relative contributions of lignin increased (+7%) and that of lipids decreased (-11%) moving from the upland to the lowland position (Fig. 6b, Table S5). In the subsurface horizons, however, both $AI_{mod}$ and average molecular weight showed little changes (Fig. 6a, Table S6), while the relative abundance of lignin increased (+9%) and that of lipids decreased (-11%).

# 4 Discussion

## 4.1 Environmental parameters controlling $CO_2$ emissions

Our field data support our hypothesis that reducing conditions under flooded conditions inhibit microbial respiration and thus reduce $CO_2$ emissions in the lowland position. Indeed, seasonal $CO_2$ emissions in the lowland positions were strongly correlated with VMC, water table depth, and $E_h$ (Fig. 3). Conversely, in upland positions where oxygen limitations are not limiting, soil temperature was found to be the best predictor variable for $CO_2$ emissions (Fig. 3a, Table 2). Our results further indicated that the impact of seasonal drainage of the lowland soils on $CO_2$ effluxes is limited by temperature effects (Fig. 2, Table 2). Oxygenation in other seasonally flooded soils usually results in increases in $CO_2$ effluxes due to enhanced aerobic microbial respiration (Laine et al., 1996; Krauss and Whitbeck, 2012). Although our lowland soils became oxygenated in the non-growing season due to the water table drop, we observed near equal $CO_2$ emissions from the three landscape positions during that time period (Table 1, Fig. 2a). A possible explanation for this convergence in $CO_2$ emissions is that oxygenation coincides with the low seasonal temperatures during the non-growing season (-1.7 to 10 degrees Celsius) which inhibit microbial activity (Lloyd and Taylor, 1994). In other words, even when seasonal drainage oxygenates the lowland soils, allowing for aerobic microbial respiration to occur (Sutton-Grier et al., 2011), $CO_2$ efflux in these seasonal wetlands still remains suppressed due to cold temperatures. In contrast, $CO_2$ fluxes along a forested wetland gradient in the Southeastern US showed a lesser seasonal convergence (Krauss and Whitbeck, 2012), as the average annual air temperature of this study site is 7 degrees Celcius above the annual average in the Northeastern US. It remains to be seen if higher temperatures during the non-growing season, as expected throughout the Northeastern US with climate change (Karmalkar and Bradley, 2017), disproportionally increase microbial respiration (and potentially C loss) from these soils.

## 4.2 Contrasting impacts of roots, mineral composition and redox on C content along the upland-to-lowland transect

C content in the lowland surface soils were nearly four-times greater than in the upland surface soils (Table 3), which was most likely caused by lower microbial respiration rates (Fig. 2a) rather than differences in C inputs. Given the proximity of our three landscape positions and minor changes in elevation, aboveground litter inputs can be considered equal across the transect. Moreover, if belowground C inputs were responsible for the greater C content, we would expect root biomass to be higher in lowland than in upland positions. In fact, the opposite was the case (Table 3). Our linear mixed effects model further

showed that C content in the surface soils were inversely related to $E_h$ across the upland-to-lowland transect (Fig. 4b). In other words, low $E_h$ values (i.e., oxygen availability) coincided with high C content in the surface soil, an observation consistent with findings by Hall and Silver (2015) in tropical surface soils. Hence, greater C content in lowland surface soil horizons are likely due to oxygen limitations rather than greater above or belowground C inputs.

Surprisingly, this relationship did not hold true in the subsurface horizons, where our linear mixed effects model showed that $E_h$ failed to predict C content across the transect (Fig. 4b, Table S3). Lower C content in lowland as compared to adjacent upland subsurface soils (Table 3) were likely a consequence of differences in root biomass; a difference that can be attributed to restricted root growth under oxygen limitations (Tokarz and Urban, 2015). With roots recognized as primary C inputs belowground, especially in the subsoil (Rasse et al., 2005), the lack of root-derived C may explain the low C content in

deeper lowland horizons. With limited C inputs at depth, microbial oxygen consumption resulting from heterotrophic respiration may not be sufficient to cause prolonged oxygen limitations (Keiluweit et al., 2016). These results suggest that the effect of oxygen limitations on C accumulation in seasonally flooded mineral soils may be most pronounced in C-rich surface soils, and less so in C-depleted subsurface soils.

          Contrasting trends between upland and lowland soils were also found for the relationship between C content and the

presence of reactive Fe and Al phases, which are known to contribute to C accumulation (Wagai and Mayer, 2007). The amount of $Fe_o$ was significantly lower (Table 3) and had significantly less power to predict C content (Fig. 4a-b, Table 4) in lowland soils than in the upland soils. The diminished importance of $Fe_o$ in C accumulation in our seasonally flooded lowland soils is consistent with the loss of reactive Fe phases observed in forest soils (Fiedler and Kalbitz, 2003; Zhao et al., 2017) and rice paddy soils (Favre et al., 2002; Kögel-Knabner et al., 2010; Hanke et al., 2014). High concentrations of organic C in rice paddy

surface soils drives the reduction and dissolution of redox-active minerals, such as Fe(III) oxides, which is subsequently translocated vertically down the soil profile (Kögel-Knabner et al., 2010, Chen et al., 2017). In our sites, we found a noticeable, yet insignificant, increase in $Fe_o$ contents in the lowland $C_g$-horizons (Table 3), which is likely a reflection of these vertical transport processes of soluble or colloidal Fe phases into the subsurface horizon, where they may reprecipitate during drained periods (Kögel-Knabner et al., 2010; Hanke et al., 2014).

While the predictive power of $Fe_o$ diminished across the upland-to-lowland transect, $Al_o$ became are stronger predictor of C content (Fig. 4a-b, Table 4). In contrast to Fe, Al hydroxides are not reducible to a more soluble lower oxidation state. Al hydroxides are thus more likely to accumulate in a dynamic redox environment such as our lowland soils. In fact, we found consistently higher $Al_o$ than $Fe_o$ contents in the lowland soils (Table 3). In similarly dynamic forested floodplain environments,

C content was also found to be more strongly correlated with $Al_o$ than $Fe_o$, which was attributed to the formation of stable $Al_3^+$-OM complexes (Darke and Walbridge, 2000). High OM contents, as found in our lowland soils, have also been found to stabilize $Al_3^+$-OM complexes by inhibiting crystallization of Al into more crystalline, and less reactive Al oxides (Darke and Walbridge, 2000; Borggaard et al., 1990). Thus, C accumulation in our seasonally flooded mineral wetland soils may partly depend on non-reducible, poorly crystalline $Al_3^+$-OM complexes.

Together, these results clearly illustrate that the relative importance of roots, mineral composition and redox conditions on C storage shifts not only along the upland-to-lowland transect, but also with depth. On the one hand, C accumulation in our upland soils relied upon both root inputs and the presence of both Fe and Al phases, as previously documented. On the other hand, reducing conditions (or oxygen limitations) in the seasonally flooded lowland soils are sufficient to cause C accumulation in the surface horizon relative to upland soils, despite lower root C inputs and lower

abundance of reactive Fe and Al phases. In contrast, subsurface horizons in seasonally flooded lowlands had much lower C content than in the uplands; here C accumulation appears to be owed to non-reducible $Al_3^+$-OM complexes, but is limited by the lack of root C inputs belowground and the absence of reactive Fe phases.

**4.3 Divergent controls on organic matter composition in seasonally flooded surface and subsurface soils**

We hypothesized that anaerobic periods during seasonal flooding limit the depolymerization of larger

macromolecular compounds and/or the microbial respiration of chemically-reduced OM in the lowland soils. Conversely, we expected the upland positions to contain smaller and chemically more oxidized OM as a result of consistently largely aerobic conditions. While prior studies have primarily focused on total C in surface soils (Hall and Silver, 2015), subsurface soils (Olshansky et al., 2018), or DOM (Rouwane et al., 2018), this work represents the first examination of the depth-resolved chemical characteristics of C composition across upland-to-lowland transitions. Analysis of the composition of solid-phase

and water-extractable C supported our predictions of a greater abundance of lignin-rich, higher-molecular weight, chemically-

reduced OM in the lowland positions, but only in the surface horizons (Fig. 5, Fig. 6). In the surface horizons, solid-phase OM across the upland-to-lowland transects became more enriched in relatively reduced aromatic and aliphatic C and depleted in relatively oxidized carboxylic C, causing the average oxidation state to gradually decrease (Fig. 5a). Along the same transect, the average molecular weight, aromaticity and contribution from lignin compounds in water extractable OM increased (Fig. 6a-b). The selective preservation of chemically-reduced, high-molecular weight OM in the lowland surface soils confirms our assertion above that comparatively low $CO_2$ fluxes and high C accumulation in the lowland surface horizons is controlled by oxygen limitations.

Contrary to our expectation, subsurface soils showed the reverse trend. Solid-phase C became more oxidized along the upland-to-lowland transect (Fig. 5). Enhanced C oxidation in seasonally flooded soils is consistent with reports by Olshanksy et al., (2018), who showed that wet-dry cycles increased the interactions between more oxidized OM constituents (i.e. carboxylic C) and reactive soil minerals. It is also well known that subsurface soils in seasonally flooded mineral soils receive significantly greater amounts of dissolved OM leaching down from the surface horizons compared to upland soils (Fiedler and Kalbitz, 2003). In forest soils, dissolved OM leachates have been shown to consist of partially-oxidized aromatic acids, presumably derived from lignin decomposition at the surface, that preferentially associate with reactive Fe phases in the subsurface (Kramer et al., 2012). As noted above, while lowland horizons showed an overall decline in $Fe_o$ and $Al_o$ contents relative to the upland position (Table), a modest uptick in $Fe_o$ content was observed in the lowland subsurface ($C_g$) horizon. A study of rice paddy soils showed that reductively dissolved Fe (hydr)oxide coatings on vermiculite clay surfaces were re-precipitated on mineral surfaces upon re-oxygenation during water table drawdown (Favre et al., 2002). Therefore, it is likely that seasonally reduced Fe (hydr)oxides are transported down the soil profile and then re-precipitated in the subsurface soil horizon during water table drawdown (Kögel-Knabner et al., 2010)—trapping dissolved, partially-oxidized, lignin-derived OM also leaching down the profile and so resulted in the accumulation of relatively-oxidized OM.

Additionally, changes in C oxidation state in the subsurface may be driven by variations in root C inputs along the upland-to-lowland transect (Table 3). Root C inputs are composed of chemically reduced aliphatic (e.g. suberin and cutin) and aromatic compounds (e.g. lignin and tannins) (Spielvogel et al., 2014). With root biomass in upland positions being noticeably higher, such root-derived inputs may have resulted in greater contributions of chemically-reduced OM (Liang and Balser,

2008). In contrast, the lowland subsurface soils were nearly void of roots (Table 3). If OM in lowland subsurface soils predominantly stems from dissolved, partially-oxidized OM leaching down the profile, as discussed above, the lack of root-derived, reduced OM compounds may result in an average C oxidation state that is relatively more oxidized.

**4.4 Interplay among redox controls, mineral protection and vegetation dynamics will determine climate change response of C storage in seasonally flooded mineral soils**

Our results indicate that oxygen limitations account for the significant C accumulation in surface horizons of seasonally flooded mineral soils. The Northeastern US is the fastest warming region in the contiguous US, with winter temperatures rising at a higher rate than summer temperatures (Karmalkar and Bradley, 2017). Similarly, precipitation is expected to increase—an increase that is predicted to occur almost exclusively in the winter months (Karmalkar and Bradley, 2017). It is thus assumed that the duration and extent of flooding within similar wetland systems will change throughout the Northeastern US (Brooks, 2005). In the summer months, we would assume that increasing temperature will cause greater evapotranspiration if precipitation remains roughly the same. Consequently, summer drainage of seasonally flooded mineral soils will likely become more pronounced. The resulting oxygenation of the surface soils may lift metabolic constraints on OM depolymerization and respiration, and is thus likely to cause soil C loss and greater $CO_2$ emissions. In the winter months, however, when both temperature and precipitation are expected to increase (Karmalkar and Bradley, 2017), seasonally flooded mineral soils around the Northeastern US will most likely remain flooded. Yet the increase in temperature may help overcome temperature limitations that we found to control emissions in winter months. To assess how the total C balance within seasonally flooded mineral wetlands may respond to climate change in the Northeastern US, it appears pertinent to explore how warmer winter temperatures affect anaerobic metabolic rates under fully saturated conditions over the winter months, and whether they have the potential to increase $CO_2$ or $CH_4$ emissions. Additionally, recent studies suggest that colonization by deeper-rooting upland plants will offset some of the C loss upon drainage of former wetlands through additional C inputs (Laiho, 2006; Mueller et al., 2016). In our system, seasonal flooding over pedologic time scales has resulted in an overall loss of reactive metal phases. This result suggests a limited capacity for new C inputs to associate with reactive Fe or Al phases, and, consequently, a low potential to offset the losses of anaerobically protected C upon drainage in the short-term. The

question whether increased root growth, and associated root-driven weathering of primary minerals, might also increase the abundance of reactive metal phases (Yu et al., 2017), and thus the potential for increased C storage in the long-term, warrants future research.

**5 Conclusions**

Our examination of $CO_2$ emissions, C content, and organic matter composition across six-replicated upland-to-lowland transects yielded important insights into the controls on C cycling in seasonally flooded mineral soils. Importantly, we see distinctly different mechanisms controlling C content and organic matter composition in surface versus subsurface soils, which sharply contrasts those governing the upland system. While $Fe_o$ and $Al_o$ predicted C content at the upland sites, $E_h$ and $Al_o$ best explained the significantly larger C accumulation in lowland soils. In spite of seasonal re-oxygenation of the surface horizons, periodic flooding (and the associated oxygen limitations) imposed sufficient metabolic constraints on depolymerization to cause the accumulation of plant-derived, aromatic, high-molecular weight OM in surface soils. In the subsurface horizons of seasonally flooded soils, anaerobic protection of C appears to be less important. C accumulation was low and primarily correlated with $Al_o$, and the OM preserved at depth was relatively oxidized. The fact that anaerobic periods during flooding restricted root growth and caused a relative depletion of Fe(III) oxides in the subsurface soil suggests that the lack of root C inputs and reactive metal phases are primarily responsible for the low subsurface C accumulation. Our findings suggest that anaerobically protected C in seasonally flooded surface soils may be particularly vulnerable to increased frequency of droughts. The extent to which associated C losses from surface soils may be compensated by upland plant encroachment and deeper root growth warrants further research.

**Acknowledgments**

The authors thank one A Thompson and three anonymous reviewers for their constructive comments. We thank J Van Sickle for his assistance with the linear mixed effects model and P Megonigal for fruitful discussions of the results. The authors are grateful to T Regier, J Dynes, and Z Arthur for their assistance and support at the CLS SGM beamline 11ID-1. Research described in this paper was performed at the Canadian Light Source, which is supported by the Canada Foundation for

Innovation, Natural Sciences and Engineering Research Council of Canada, the University of Saskatchewan, the Government of Saskatchewan, Western Economic Diversification Canada, the National Research Council Canada, and the Canadian Institutes of Health Research.. The research was performed using EMSL, a DOE Office of Science User Facility sponsored by the Office of Biological and Environmental Research. Work conducted by REL was supported by the USDA National Institute of Food and Agriculture (Hatch project 1007650). This work was supported by the Department of Energy, Office of Biological and Environmental Research, Subsurface Biosphere Research program (Award no. DE-SC0016544)

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

**Figure Captions**

**Figure 1. Illustration of upland-to-lowland transects in forested seasonally flooded mineral soils used for this study.** (a) Approximate distances and elevation change between landscape positions along the transects as well as the horizons sampled within each position. Approximate seasonal high and low water table depths are indicated by dashed lines. Example of (b) flooded and (c) drained seasonal wetland with marked upland (U), transition (T) and lowland (L) positions.

**Figure 2. $CO_2$ efflux, water table, moisture and redox dynamics along upland-to-lowland transects.** Mean monthly (a) soil $CO_2$ efflux, (b) water table depths, (c) volumetric moisture contents, and (d) depth-resolved redox potentials for the three landscape positions; upland, transition and lowland. Redox potentials are standardized from a calomel to a standard hydrogen electrode. Data are the means of measurements along upland-to-lowland transects in six replicate wetlands. Missing data points for the upland position in panel (b) are due to the water table dropping below the measuring gauge installed at a depth of 50 cm and is denoted with asterisks (*).

**Figure 3. Pairwise regressions between soil $CO_2$ efflux and soil temperature, water table depth, moisture content and redox potential.** Monthly averages for each environmental variable, recorded in six replicate upland-to-lowland transects over a full year, were combined for regression analyses. Regression analyses were conducted for both growing (red-scale markers) and non-growing season (blue-scale markers). (a) Relationship between soil temperature at 10 cm depth and soil respiration modelled using the Arrhenius equation. (b) Linear regressions of water table depths against $CO_2$ efflux. Water table depths less than zero are below soil surface; depths greater than zero are above soil surface. (c) Linear regression of volumetric moisture contents at 10 cm depth plotted against $CO_2$ efflux. (d) Linear regressions of soil redox potentials at 15 cm depth plotted against $CO_2$ efflux. Growing season (GS) and non-growing season (NGS) fits are shown for each regression.

**Figure 4. Fixed effect parameters predicting total C in linear mixed effects models.** (a) F-values of fixed effects for $Al_o$, $Fe_o$, clay, root biomass, and mean growing season $E_h$ in each landscape position. (b) F-values of fixed effects for $Al_o$, $Fe_o$, clay, root biomass, and mean growing season $E_h$ in the different horizons.

**Figure 5. C (1s) NEXAFS analyses of solid-phase OM chemistry across upland-to-lowland transects.** (a) NEXAFS spectra from six replicate wetlands (grey), plotted for each landscape position and depth, with the resulting mean spectra plotted (black). Peaks of particular interest are carboxylic C (285.35 eV), aliphatic C (287.20 eV), and aromatic C (285.03 eV) denoted by dotted vertical lines. (b) Average carboxyl-to-aromatic C ( 285.35 eV/285.03 eV) ratios plotted for each landscape position and depth; bars are standard error of the mean of the replicates.

**Figure 6. FT-ICR-MS analysis of water-extractable OM chemistry across upland-to-lowland transects.** (a) Average relative abundances of compound classes as identified by O/C and H/C ratios in Van Krevelen plots. Grey-scale colors denote primarily plant-derived compound classes, while blue-scale compounds denote microbial-derived compound classes. (b) Average AImod values, as an index for aromaticity, and molecular weights of all detected compounds. Averages represent the mean of replicate samples from six upland-to-wetland transects; bars are standard error of the mean.

Table 1 Average cumulative $CO_2$ emissions (n = 6 ± standard error) for each landscape position across upland-to-lowland transects

| | Full year | Growing season | Non-growing season | Full year normalized to C content |
|---|---|---|---|---|
| | mol $CO_2$ m$^{-2}$ | mol $CO_2$ m$^{-2}$ | mol $CO_2$ m$^{-2}$ | mol $CO_2$ m$^{-2}$ mg C$^{-1}$ g soil$^{-1}$ 10 |
| Upland position | 54[a] ± 1.1 | 38[a] ± 1.6 | 16[a] ± 0.8 | 1.15[b] ± 0.23 |
| Transition position | 53[a] ± 0.9 | 33[a] ± 1.5 | 20[a] ± 0.9 | 1.03[a] ± 0.22 |
| Lowland position | 39[a] ± 0.8 | 24[a] ± 1.3 | 15[a] ± 0.7 | 0.39[a] ± 0.11 |

Letter designations are Tukey's honestly significance test results. Different letter designations indicate a p-value of $< 0.05$.

Table 2 Regression analysis (r) results of potential environmental variables that predict $CO_2$ emissions along a moisture gradient

| Environmental Variable | Season | Upland | Transition | Lowland |
|---|---|---|---|---|
| Soil temperature[#] | Full | 0.72*** | 0.60*** | 0.53*** |
| | GS | 0.62*** | 0.56*** | 0.45*** |
| | NGS | 0.79*** | 0.81*** | 0.69*** |
| Water Table Depth[S] | Full | -0.03 | -0.05 | -0.30** |
| | GS | -0.32** | -0.14 | -0.55*** |
| | NGS | -0.20 | -0.17 | -0.35** |
| Volumetric Moisture Content[S] | Full | 0.20* | -0.44*** | -0.32*** |
| | GS | 0.10 | -0.72*** | -0.51*** |
| | NGS | -0.10 | -0.37** | -0.37** |
| Soil Redox Potential[S] | Full | 0.10 | 0.10 | 0.01 |
| | GS | 0.05 | 0.41*** | 0.40*** |
| | NGS | 0.06 | 0.08 | 0.27* |

Full = entire year, GS = growing season, NGS = non-growing season.
[#] Arrhenius fit
[s] Linear fit
Significance codes: < 0.001 = '***', 0.01 = '**', 0.05 =' *'

Table 3 Average (n = 6 ± standard error) soil properties along the upland-to-lowland transect

| Horizon | Total Carbon (%) | C:N | Root Biomass (mg g$^{-1}$ soil) | pH | Silt + Clay (%) | Fe$_o$ (mg g$^{-1}$ soil) | Al$_o$ (mg g$^{-1}$ soil) |
|---|---|---|---|---|---|---|---|
| Upland | | | | | | | |
| A | 2.3[a] ± 0.5 | 11[a] ± 2.5 | 61[a] ± 27 | 4.98 ± 0.2 | 48[a] ± 11 | 3.6[a] ± 0.5 | 5.1[b] ± 0.8 |
| B | 1.1[ab] ± 0.3 | 13[a] ± 3.4 | 14[a] ± 3 | 5.22[a] ± 0.2 | 39[a] ± 11 | 2.4[a] ± 0.7 | 5.7[a] ± 1.9 |
| C | 0.64[a] ± 0.1 | 13[a] ± 5.3 | 6[a] ± 3 | 5.29[a] ± 0.1 | 37[a] ± 12 | 1.7[a] ± 0.4 | 3.6[a] ± 0.8 |
| Transition | | | | | | | |
| A | 3.9[ab] ± 1.5 | 14[a] ± 1.6 | 48[ab] ± 17 | 4.97[a] ± 0.2 | 51[a] ± 10 | 1.2[a] ± 0.4 | 2.5[a] ± 0.3 |
| B/C | 0.64[a] ± 0.1 | 6.4[a] ± 1.4 | 15[a] ± 6 | 5.38[a] ± 0.1 | 41[a] ± 11 | 1.5[a] ± 0.3 | 1.9[a] ± 0.3 |
| Cg | 0.36[a] ± 0.1 | 5.0[a] ± 3.8 | 3[a] ± 1 | 5.43[a] ± 0.2 | 59[a] ± 10 | 1.3[a] ± 0.3 | 1.3[a] ± 0.4 |
| Lowland | | | | | | | |
| A | 8.2[b] ± 2.4 | 16[a] ± 1.1 | 6[b] ± 2 | 4.98[a] ± 0.1 | 50[a] ± 13 | 1.5[b] ± 0.4 | 4.1[ab] ± 1.1 |
| C | 1.9[b] ± 0.5 | 13[a] ± 3.7 | 2[b] ± 0.6 | 5.29[a] ± 0.1 | 66[a] ± 9 | 1.0[a] ± 0.3 | 2.6[a] ± 0.5 |
| Cg | 0.36[b] ± 0.02 | 7.3[a] ± 4.9 | 0.7[a] ± 0.5 | 5.37[a] ± 0.1 | 70[a] ± 9 | 2.9[a] ± 0.7 | 1.0[b] ± 0.2 |

Letter designations indicate significant differences among horizons of similar depth. Comparisons within surface (A horizons), intermediate (B/C horizons), and subsurface (C/Cg horizons) horizons were determined by ANOVA followed by Tukey's HSD. Different letter designations indicate a p-value of $< 0.05$.

Table 4 Fixed effect parameters from the linear mixed models along the upland-to-lowland transects

| Variable | Degrees of freedom | Regression Coefficient ± standard error | F - value | Landscape Prob > F | Horizon Prob >F |
|---|---|---|---|---|---|
| **Upland** | | | | | |
| Root Biomass | 17 | $0.15 \pm 0.09$ | 13.31 | <0.01 | NS |
| $Fe_o$ | 17 | $0.37 \pm 0.15$ | 17.31 | <0.001 | NS |
| $Al_o$ | 17 | $0.31 \pm 0.09$ | 10.76 | <0.01 | <0.05 |
| Clay | 17 | $0.39 \pm 0.23$ | 8.56 | <0.01 | <0.05 |
| $E_h$ | 17 | $0.24 \pm 0.31$ | 2.86 | NS | <0.05 |
| **Transition** | | | | | |
| Root Biomass | 18 | $0.05 \pm 0.07$ | 21.81 | <0.001 | <0.05 |
| $Fe_o$ | 18 | $-0.05 \pm 0.08$ | 2.86 | NS | <0.0001 |
| $Al_o$ | 18 | $0.14 \pm 0.14$ | 15.57 | <0.001 | <0.001 |
| Clay | 18 | $0.24 \pm 0.07$ | 1.68 | NS | <0.0001 |
| $E_h$ | 18 | $-0.08 \pm 0.08$ | 0.32 | NS | <0.0001 |
| **Lowland** | | | | | |
| Root Biomass | 12 | $0.12 \pm 0.38$ | 11.22 | <0.01 | NS |
| $Fe_o$ | 12 | $-0.01 \pm 0.17$ | 0.45 | NS | <0.01 |
| $Al_o$ | 12 | $0.91 \pm 0.15$ | 137.36 | <0.0001 | NS |
| Clay | 12 | $-0.08 \pm 0.21$ | 0.31 | NS | <0.01 |
| $E_h$ | 12 | $-0.47 \pm 0.23$ | 0.77 | NS | <0.01 |

Model parameters with p-values > 0.05 are denoted as not-significant with the letters NS.

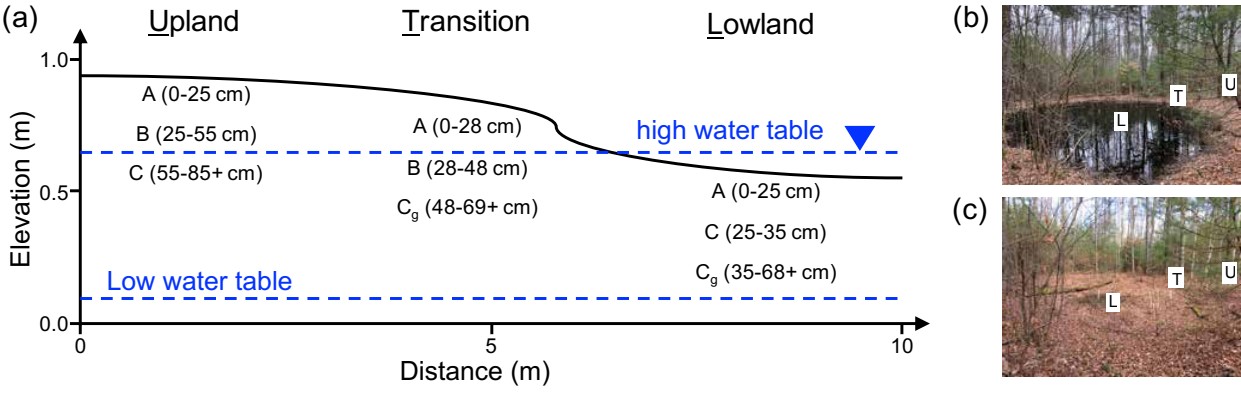

(a) Upland   Transition   Lowland

Elevation (m)

1.0

A (0-25 cm)
A (0-28 cm)
B (25-55 cm)
B (28-48 cm)
high water table
C (55-85+ cm)
$C_g$ (48-69+ cm)
0.5
A (0-25 cm)
C (25-35 cm)
Low water table
$C_g$ (35-68+ cm)
0.0

0   5   10

Distance (m)

(b)

(c)

fig01

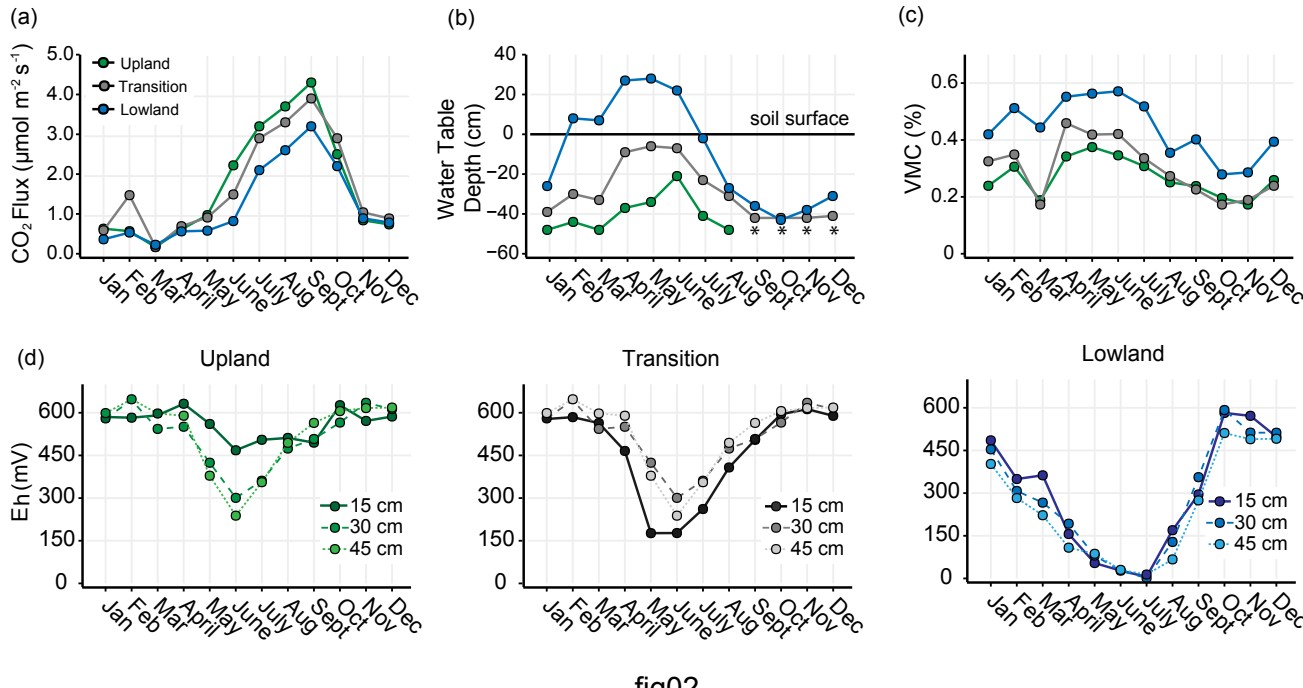

fig02

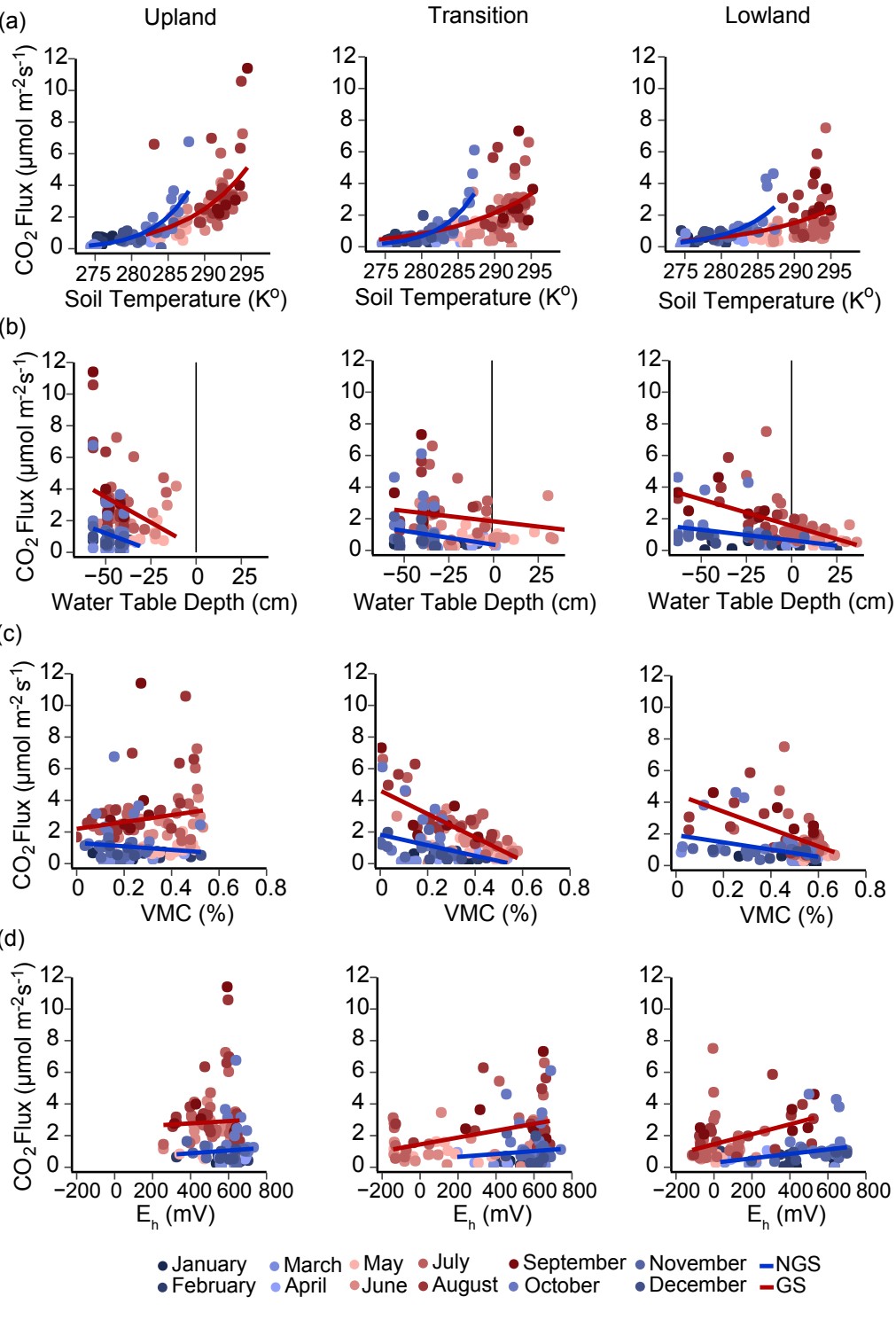

fig03

(a)

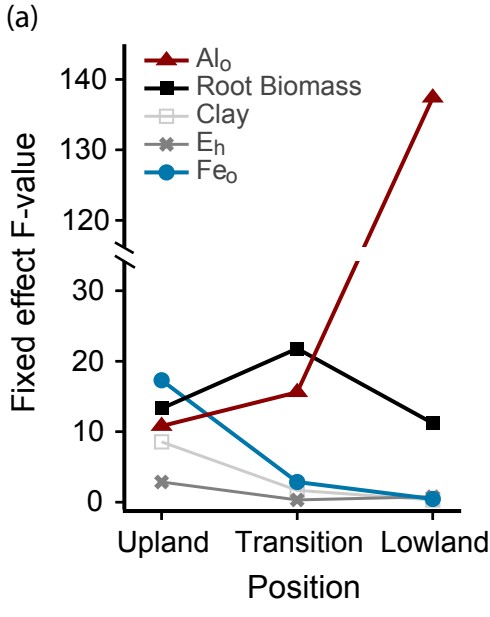

(b)

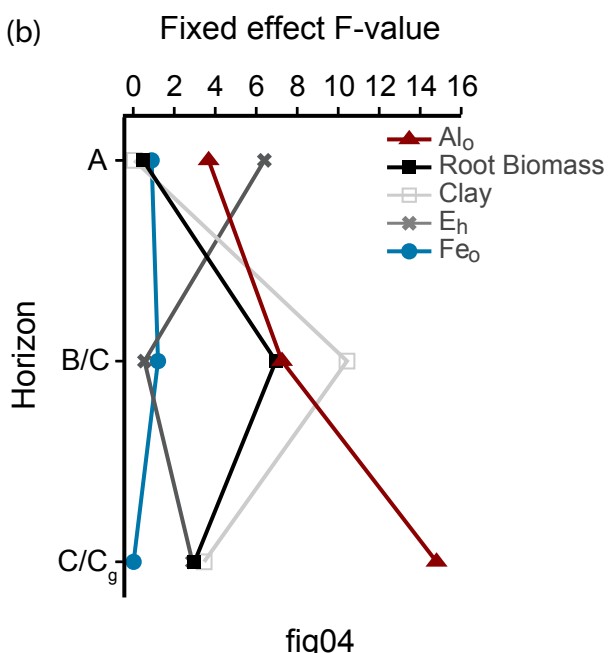

fig04

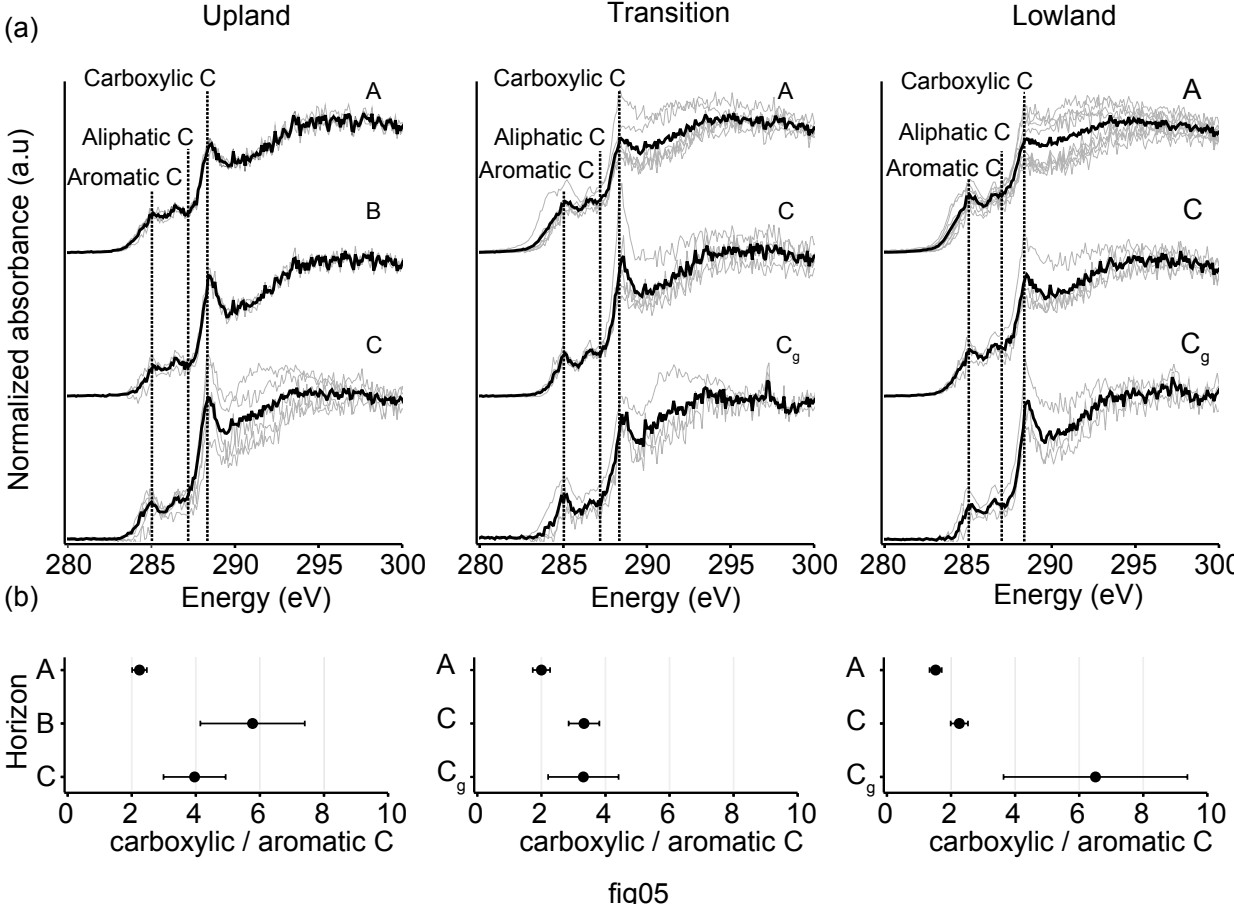

fig05

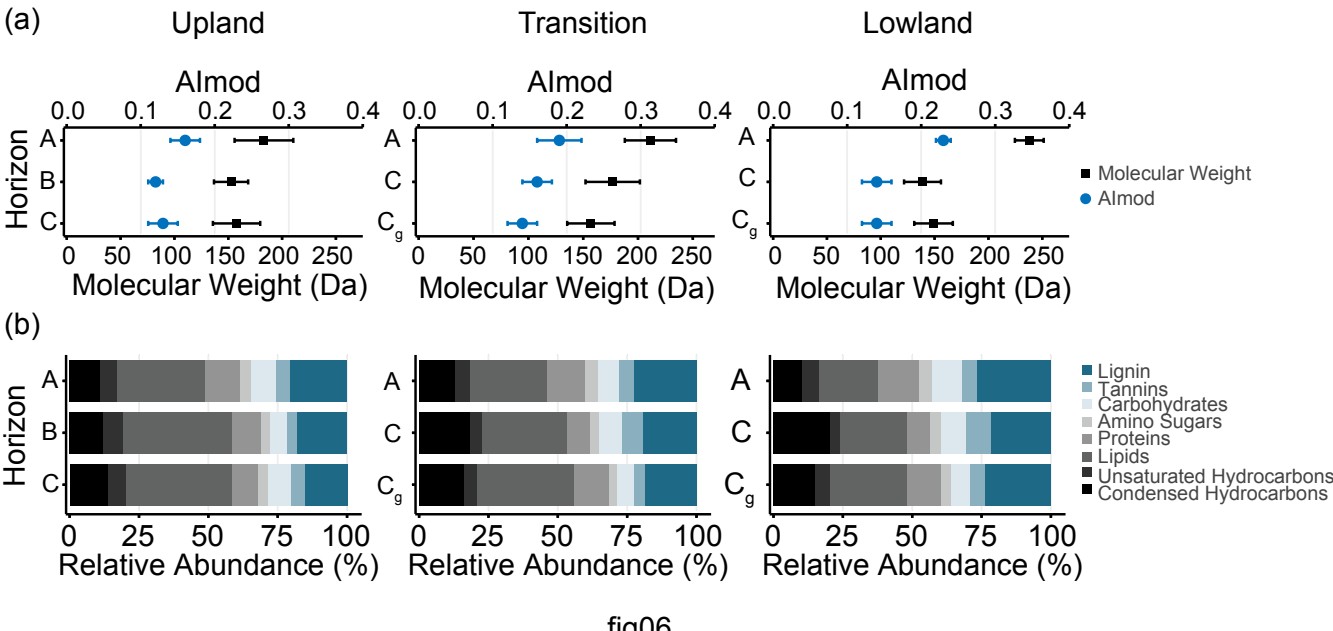

fig06