# Peer review of "Shifting Mineral and Redox Controls on Carbon Cycling in Seasonally Flooded Mineral Soils"

_Biogeosciences, 2018_

## Referee Comment (RC1) · A. Thompson (Referee) · 19 Nov 2018

Overall I think this is a valuable dataset and a well-executed study. I support its eventual publication. The surface horizon data are well established and I have just minor comments there as indicated below. However, the subsurface depth data are problematic, mostly because there are different overall soil depths in each of the sites and different horizon designations. This has led also I think to some statements that are not well supported by the statistics or that the statistics used are not well presented. For instance, reading the abstract while looking at Table 3 raises several questions if we interpret "significantly lower" to mean different statistical lowercase letter assignments, which most readers will. Much of my confusion occurs in section 3.3, where it appears in most cases the differences described are not statistically significant as shown on

Table 3, but this is not pointed out in the text. At one point here the authors refer to a Tukey test for the topsoils (although I think they misplaced the word subsoil on pg. 12, ln 2) with a p value < 0.01 for the transition vs. lowland, but the those share a lowercase letter assignment in Table 3, which suggests they would not have a p value < 0.01. Correcting these presentation or interpretation issues is critical. Assuming the stats letter values are correct, I think this could be resolved by looking at C stocks rather than C concentrations at depth and backing off on some of the subsurface interpretations that are not fully supported by the stats. If the authors have data bined at finer depth intervals, that might help clarify things as well, but if not I suggest using C stock down to 68 cm, in which case one could compare equally across all the sites. One could examine surface C stock (0- 25 cm) and then a subsurface C stock value (25 – 68cm). Outside of this major issue, I think the paper has a lot of promise and the combination of field $CO_2$ data and molecular-scale carbon chemistry is exciting.

Abstract I read and reviewed the abstract without looking at any other parts of the MS to mimic a reader looking at the abstract on-line. Read alone, I am not clear on the findings and implications and thus the abstract needs to be clarified. I give a couple of specifics below in the line edits, but I encourage the authors to have someone unfamiliar with the study read the abstract alone after revision.

Ln 14: Is it really true that this is largely unknown? If this is just for seasonally flooded mineral soils (compared to wetlands in general), then this point escaped me on the first read. Perhaps it was the shift from "seasonally flooded soils" in the previous sentence to "seasonally flooded mineral soils" in this sentence. Use one term and stick with in, especially in the abstract where space is tight. Ln 16: Need to specify here that the lowlands are periodically flooded and the uplands are not—if that is indeed the case. I am assuming that, but one could have uplands that are also periodically flooded due to high rainfall and perched water tables. Ln 17: This sentence is hard to follow. I read it twice and was still not sure what it was saying, where C was higher? I suggest "We found the lowlands had lower $CO_2$ effluxes than the uplands. Lowland surface

soils (0-20 cm. . .or whatever it is—also could give A or B or O classification) had higher C concentrations a higher abundance. . .than the uplands." Ln 20: Here I was confused again by subsoils slipping in there. I think you need to be much more upfront about this distinction as it is one of the main points of the abstact. At the end you also start to talk about C stocks (depth integrated concentrations), which would take into account bulk density. Consider discussing that here instead of concentration? Ln23: It is not clear what non-reducible Al phases are being relied on for here? I assume mineral protection, but best not to have readers assuming in the abstract. Ln 24-25: The three reasons given for why you see more C in the topsoils than the subsoils are not supported in the abstract by any data. Either include this data upfront (i.e., lowland had low/zero O2, whereas uplands had O2 above X%; also data on roots and Fe presence/abundance) or you could simply state that these C findings correlated with O2, roots and Fe, implying the data is in the paper, but not fully presented. What you are asking the reader to do here is accept this statement without any sense that it is supported by data in this paper and that is not comfortable to many readers (and me I suppose). Ln26: Again, without O2 data or mineral protection data, how could you conclude this. I assume it is in the rest of the paper. . .but I have not read that yet if I am most readers.

Introduction The introduction does a nice job setting the stage although I suggest line edits below. Pg. 2 Ln 14-15: Revise for clarity. Ln 16: Maybe not "model ecosystems" but essential "endmembers". Ln 19: This is an "endash" and you want an "emdash" here. A longer dash, that should not have spaces around it. On MS-Word you hit dash twice between words without adding spaces and word turns it into an emdash. Do this elsewhere in the text. Ln24: "seasonal wetlands" or seasonally flooded mineral wetlands, choose one term and stick with it through-out the MS.

Pg. 3 Ln 5: "catalyze" Ln 10: instead of chemically-reduced, "lower valance" would be more precise. Ln 17: ", but the impact of roots on soil C. . ." Ln 18: "growth due to low DO (Day. . .)" Ln 25: "distribution of high surface area minerals that are excellent

sorbents for C in soils"

Pg 4 I point out three of our recent papers that are highly relevant to this introduction/discussion, but which were not likely available when this was drafted. Ln 10: See Chen et al 2018 ES&T and Chen and Thompson 2018 ES&T on these topics Ln 17: See Barcellos et al 2018 Soil Systems on this topic Ln 25: "measurements of soil. . ."

Methods Well done, except that more description of the stats used are required potentially to clarify issues I raise above and below with regard to Table 3 lowercase lettering.

Results Main issue in this section is the depth that is considered 'subsurface'. How does one determine this for soils with different depths or thicknesses? Normally, this doesn't matter, but in this case the authors are making a key argument about the C and Fe interactions and chemistry "at depth". Examining the C horizons, total C is actually higher in the lowland than in the upland and this would be true even if we examined C concentrations at 25 cm across the sites. If we go deeper, then the Cg of the transition and the lowland are equally low and the upland is higher, but not statistically higher based on the lower case letter assignments. The same is true in inverse for the lowest depth for Fe-o, it is highest at the lowland, but this is not significant from the other sites. This makes statements like "C concentrations were significantly lower in the lowland than in the upland subsoils", which is in the abstract, incorrect based on the authors' assignment of letters (see Table 3). Pg. 11 Ln 9: Assuming that Feb – June is the wet period, but you should tell readers that explicitly. Ln 11: "significantly lower than in. . ." Ln 13: "season lowland VMC. . ." Ln 21: Maybe it would be helpful to calculate the EH7 values here so that these could be compared with other studies and compared between the surface and subsurface horizons. Ln 23: change "mineralogy"—which is the study of minerals—to "mineral composition". Do this elsewhere as well. Pg. 12 Ln 1: The data are more complex than this statement suggests. Please revise. Ln 2: Do you mean topsoil here???? Because actually it is over 8 times the subsoil, but according to the letters, the lowland and transition topsoil are equal within error. Ln 4: although this was not statistically significant, correct? I suggest adding that information. Ln 7:

[Figure]

Although again this was not statistically significant, right? Tell the reader that. Ln 9: True, except in the upland, right (Table 3 indicates it is not significant). Ln 12: Change 'determine' to 'predict' Ln 18: "concentrations decreased along..." Ln 20: Change 'identify' to 'predict' Ln 24: Maybe not Eh, but likely O2, right? Ln 25: Change 'effect' to 'influence' Pg. 13 Ln 17: "across the upland to lowland transect..." Ln 18: "...(-11%) moving from upland to lowland." Fig. 2: Symbols are hard to tell from one another. Consider using squares, triangles and circles. Cool could help too since other figures are in color. Fig. 2: Are the Eh values on these graphs corrected for pH? To allow comparisons between the depths/sites?

Discussion

Pg. 13 Ln 22: Change 'demonstrate' to 'suggest' Ln 23: "...transects, but exhibit potentially inverse trends in the subsurface." Pg. 14 Ln 1-2: delete sentence. Ln 4: "Our field data support our hypothesis that reducing..." Ln 16-20: Clarify this section. Ln 24: Note that the figure shows topography that is not flat. Pg. 15 Ln 7: Consider using C stocks instead of concentration, which would help get around the depth issue. Pg. 17 Ln 23: OK, but Cg in the lowland is 2nd highest across ALL sites/depths, so this statement doesn't ring fully true for me.

Conclusions Ln 10: change 'related to' to 'correlated with' Ln 12: But, again what about Fe-o in the lowland Cg?????

References (1) Chen, C.; Thompson, A., Ferrous iron oxidation under varying pO2 levels: The effect of Fe (III)/Al (III) oxide minerals and organic matter. Env. Sci. Technol. 2018, 52, 597-606.

(2) Chen, C.; Meile, C.; Wilmoth, J.; Barcellos, D.; Thompson, A., Influence of po2 on iron redox cycling and anaerobic organic carbon mineralization in a humid tropical forest soil. Env. Sci. Technol. 2018, 52, 7709-7719.

(3) Barcellos, D.; O'Connell, C.; Silver, W.; Meile, C.; Thompson, A., Hot spots and

hot moments of soil moisture explain fluctuations in iron and carbon cycling in a humid tropical forest soil. Soil Systems 2018, 2.

---

## Referee Comment (RC2) · Anonymous Referee #2 · 7 Dec 2018

LaCroix et al. report findings on C storage and changes in the physical-geochemical composition of soils minerals and redox conditions under the seasonal flooding soils. Nevertheless, in different part of the manuscript several strongly weak points have been identified that must be addressed from the authors. Moreover, I do not find that results provided insight into the mechanism on C storage and the changes in the shifting minerals and other critical factors.

1. Synchrotron-based X-ray analyses and FT-ICR-MS analyses. There are plenty of literatures on these methods, as a reviewer also a reader, I suggest simplify these parts and move the detail descriptions into SI.

2. Results. In figure 2, the symbols are too small and similar to be recognized, while and the resolution are lower.

[Figure]

3. In 3.3, compared with Feo and Alo, the authors haven't presented the detailed data of Fed and Ald, which are more sensitive to the changing of environmental factors. Meanwhile, the soil iron cycling is sensitive to the seasonal flooding, the recrystallization processes of iron oxides as well as aluminum oxides during the shifting of seasonal flooding soils are critical factors to the variation of the iron/aluminum species, which are further controlled the reactivity of iron/aluminum species in soil environment.

4. In 3.4, carboxylic/aromatic C ratios is a suitable indicator to present the different of oxidation degrees, however, from fig 5b, it's inaccurate to describe the increase trends of those values above in C horizons. It doesn't present significant different in fig 5b.

5. In discussion part, the authors just repeated the obtained results described in results part in another similar way, lacking of further discussion around the mechanism among C storage and the changes in the shifting minerals and other critical factors. As a reader, I find it's hard to get new information in this important parts.

6. Further experiments should be designed and conducted to illustrate the mechanism on soil chemical-physical properties and C storage.

---

## Author Comment (AC1) · 15 Jan 2019

**RESPONSE TO REVIEWERS' COMMENTS**

Regular font:   original comment by the reviewer
*Italized text:*          response by the authors
*"Italicized quotes":*   revised text segments

**Response to Comments by Referee #1**

*Comment:* Overall, I think this is a valuable dataset and a well-executed study. I support its eventual publication. The surface horizon data are well established, and I have just minor comments there as indicated below. However, the subsurface depth data are problematic, mostly because there are different overall soil depths in each of the sites and different horizon designations. This has led also I think to some statements that are not well supported by the statistics or that the statistics used are not well presented. For instance, reading the abstract while looking at Table 3 raises several questions if we interpret "significantly lower" to mean different statistical lowercase letter assignments, which most readers will. Much of my confusion occurs in section 3.3, where it appears in most cases the differences described are not statistically significant as shown on Table 3, but this is not pointed out in the text. At one point here the authors refer to a Tukey test for the topsoils (although I think they misplaced the word subsoil on pg. 12, ln 2) with a p value < 0.01 for the transition vs. lowland, but the those share a lowercase letter assignment in Table 3, which suggests they would not have a p value < 0.01. Correcting these presentation or interpretation issues is critical.

*Authors' response:  We thank Dr. Thompson for these extremely thoughtful and constructive comments.  We took the following steps to assure that the statistical tests and the resulting estimates of significance are reported correctly.*

*First, we emphasize that the statistical analyses were done on square root transformed data, but the data provided in Table 3 is the non-transformed observational data. We have clarified this in the methods section on Page 10: " Statistical analyses were conducted on square root transformed data when assumptions of normal distribution were not met, although non-transformed observational values are reported within the text (Table 3)."*

*Second, we clarified our statements regarding the significant differences in C contents in top and subsoils across the transect.  We believe that the confusion arose from the fact that we made statements in the text (both abstract and main body) that were not reflected in the statistics reported in Table 3. The results presented in Table 3 are based on a very conservative approach (Tukey's honesty significance) to determine significance of the means. This test yielded the letter designations reported in Table 3. The ANOVA model uses and distributes the variance for the entire dataset and the 36 individual comparisons. In essence, the high variability, particularly in the A-horizons, is distributed across the entire model. Consequently, an ANOVA model taking all 36 horizons into account yielded no significant differences, even though the means in some cases seemed different from an ecological perspective (2.3 versus 8.5% in the A horizons and 0.4 versus 0.6% in C/Cg horizons). Because we were primarily concerned with comparing C contents across individual horizons along the transect (e.g., comparing A horizons across upland, transition and lowland position), we had decided to conduct additional ANOVAs on a horizon-basis. Those tests indicated that both A and C horizons in upland and lowland positions are significantly different.  We had presented the results from this analysis, including the p*

*values indicating significant differences among the horizons, in the abstract (Ln: 20-22) and results section (Pg. 11 Ln:21-25 ). In the revised version, we explain these additional comparisons in the text and added the full results table to the SI (Table S7). Additionally, Table 3 was updated to indicate the significance based on horizon groups. We hope that these measures will clarify this issue.*

Assuming the stats letter values are correct, I think this could be resolved by looking at C stocks rather than C concentrations at depth and backing off on some of the subsurface interpretations that are not fully supported by the stats.

*Authors' response: We thank the reviewer for this suggestion. We believe the above explanation of the statistical approach taken to support our interpretations resolves the confusion about the stats letters.*

If the authors have data bined at finer depth intervals, that might help clarify things as well, but if not I suggest using C stock down to 68 cm, in which case one could compare equally across all the sites. One could examine surface C stock (0- 25 cm) and then a subsurface C stock value (25 – 68cm).  Outside of this major issue, I think the paper has a lot of promise and the combination of field CO2 data and molecular-scale carbon chemistry is exciting.

*Authors' response: We thank the reviewer for this suggestion, but believe that the measures taken above to clarify our statistical approach sufficiently address this concern.*

Abstract

Comment: I read and reviewed the abstract without looking at any other parts of the MS to mimic a reader looking at the abstract on-line. Read alone, I am not clear on the findings and implications and thus the abstract needs to be clarified. I give a couple of specifics below in the line edits, but I encourage the authors to have someone unfamiliar with the study read the abstract alone after revision.

*Authors' response: We revised the abstract in response to the specific comments provided below. Most importantly, we ore explicitly included the data that allowed to arrive at our conclusions.*

Comment: Ln 14: Is it really true that this is largely unknown? If this is just for seasonally flooded mineral soils (compared to wetlands in general), then this point escaped me on the first read. Perhaps it was the shift from "seasonally flooded soils" in the previous sentence to "seasonally flooded mineral soils" in this sentence. Use one term and stick with in, especially in the abstract where space is tight.

*Authors' response: We agree that the terminology used should be more consistent as 'seasonally flooded mineral soils'. The abstract and manuscript will be updated accordingly. These ecosystems (seasonally flooded mineral soils; i.e. vernal pools, ephemeral ponds, etc.) have little information published research focused soil C cycling, especially with depth.*

Authors' changes: *Ln13-15: "Among wetlands, seasonally flooded mineral soils are likely the most vulnerable to increased severity and duration of droughts in response to climate change.*

*Yet, the relative influence of associated changes in oxygen limitations, root dynamics, and mineral protection on C cycling in seasonally flooded mineral soils is largely unknown."*

Comment: Ln 16: Need to specify here that the lowlands are periodically flooded and the uplands are not if that is indeed the case. I am assuming that, but one could have uplands that are also periodically flooded due to high rainfall and perched water tables.

*Authors' response: This point has been clarified in the abstract. The lowlands are in fact seasonally flooded whereas the transition and upland position are not.*

*Authors' changes: Ln15-18: "To address this knowledge gap, we combined seasonal monitoring of soil moisture, redox potential, and $CO_2$ efflux with a characterization of root biomass, C quantity, and mineral and organic matter composition in both top- and subsoils along upland-to-lowland transects in a temperate deciduous forest. Specifically, we contrasted mineral soils in lowland positions that experience seasonal flooding with adjacent upland soils that do not."*

Comment: Ln 17: This sentence is hard to follow. I read it twice and was still not sure what it was saying, where C was higher? I suggest "We found the lowlands had lower CO2 effluxes than the uplands. Lowland surface soils (0-20 cm...or whatever it isâ˘T˘also could give A or B or O classification) had higher C concentrations a higher abundance. . .than the uplands."

*Authors' response: We revised the sentence as suggested.*

*Authors' changes: Ln18-19: "We found the lowland soils had lower $CO_2$ effluxes than the upland soils. Lowland surface soils (A-horizons) had higher C concentrations and a higher abundance of high molecular weight and chemically reduced organic compounds than the uplands."*

Comment: Ln 20: Here I was confused again by subsoils slipping in there. I think you need to be much more upfront about this distinction as it is one of the main points of the abstract. At the end you also start to talk about C stocks (depth integrated concentrations), which would take into account bulk density. Consider discussing that here instead of concentration?

*Authors' response: We revised the abstract to more clearly distinguish subsoil from topsoil results as well as C stocks from C concentration.*

*Authors' changes: Ln20-21: "These results indicate that selective preservation of organic compounds during anaerobic periods caused C accumulation in seasonally flooded surface soils (A horizons). In contrast, seasonally flooded subsoils ($C_g$-horizon) had lower C concentrations."*

Comment: Ln23: It is not clear what non-reducible Al phases are being relied on for here? I assume mineral protection, but best not to have readers assuming in the abstract.

*Authors' response: This statement was implying that 'non-redox active' Al phases, measured as ammonium oxalate extractable Al, are more strongly correlated with C than ammonium oxalate extractable Fe, which we considered "redox active". However, we agree this sentence is confusing/misleading and have revised it.*

*Author's changes*: Ln 23-24 : *"C concentrations in the subsoils were strongly correlated with ammonium oxalate extractable Al phases rather than redox potential."*

Comment: Ln 24-25: The three reasons given for why you see more C in the topsoils than the subsoils are not supported in the abstract by any data. Either include this data upfront (i.e., lowland had low/zero O2, whereas uplands had O2 above X%; also data on roots and Fe presence/abundance) or you could simply state that these C findings correlated with O2, roots and Fe, implying the data is in the paper, but not fully presented. What you are asking the reader to do here is accept this statement without any sense that it is supported by data in this paper and that is not comfortable to many readers (and me I suppose).

*Authors' response:* *We recognize that the statements within the abstract are not presented with the data to support these claims. We have revised this statement to reflect that there were correlations found between C and roots, Fe and redox potential and the supporting data is found in the main manuscript.*

*Authors' changes* Ln 24-25: *"Our linear mixed effects model demonstrated that redox potential was the better predictor for C concentrations in the topsoils (A horizon) of seasonally flooded soils, suggesting that anaerobic conditions are responsible for C accumulation."*

Comment: Ln26: Again, without O2 data or mineral protection data, how could you conclude this. I assume it is in the rest of the paper. . .but I have not read that yet if I am most readers.

*Authors' response:* *We have addressed this issue in our revision of the abstract.*

Introduction

Comment: The introduction does a nice job setting the stage although I suggest line edits below.

*Authors' response:* *We appreciate this comment and the subsequent revision suggestions for clarity.*

Pg. 2

Comment: Ln 14-15: Revise for clarity.

*Authors' response:* *The authors have revised the sentence as suggested.*

*Authors' changes:* Ln14-15: *"Seasonal wetlands can be considered early warning ecosystems (Brooks, 2005); forecasting the impacts of climate change on permanently flooded mineral wetlands."*

Comment: Ln 16: Maybe not "model ecosystems" but essential "endmembers".

*Authors' response:* *We agree that 'endmembers' is a more suitable representation of these ecosystems.*

*Authors' changes :Ln18-19: Thus, seasonally flooded wetlands represent essential endmembers to study the effects of climate change on larger permanently flooded wetland soils (Brooks, 2005).*

Comment: Ln 19: This is an "endash" and you want an "emdash" here. A longer dash, that should not have spaces around it. On MS-Word you hit dash twice between words without adding spaces and word turns it into an emdash. Do this elsewhere in the text.

*Authors' response: Thank you for noticing and correcting the endash/emdash inaccuracies. We have corrected all throughout the text.*

Comment: Ln24: "seasonal wetlands" or seasonally flooded mineral wetlands, choose one term and stick with it through-out the MS.

*Authors' response: We agree the terminology needs to be more consistent. We have revised the manuscript to be more consistent with the term 'seasonally flooded mineral soils'*

*Authors' changes: Ln24-25: "Determining the controls on C cycling within seasonally flooded mineral soils thus requires specific consideration of the fluxes and dynamics across these terrestrial-aquatic transitions."*

Pg. 3

Comment: Ln 5: "catalyze"

*Authors' response: We revised the sentence, replacing "catalyzing.." to "..which catalyze..".*

*Authors' changes: Ln4-6: "The resulting oxygen limitations inhibit the activity of oxidative enzymes which catalyze the depolymerization of higher-molecular weight OM into smaller, assimilable compounds (Megonigal et al. 2003)."*

Comment: Ln 10: instead of chemically-reduced, "lower valance" would be more precise.

*Authors' response: The authors appreciation this suggestion. To be consistent with other publications on the subject, we would like to retain the 'chemically-reduced' terminology, but also reference "oxidation state" in the revised version.*

*Authors' changes: Ln9-11: "Anaerobic conditions limit microbes to utilizing substrates that are chemically more oxidized, in turn preferentially preserving more chemically-reduced organic compounds (i.e., compounds with lower C oxidation states) in soils and sediments (Boye et al. 2017; Keiluweit et al., 2017)."*

Comment: Ln 17: ", but the impact of roots on soil C. . ."

*Authors' response: We thank the referee for this revision, which adds more clarity to the sentence.*

*Authors' changes:* Ln17-18: "Roots are the main contributors to C stocks in upland soils (Rasse et al., 2005), but the impacts of roots on soil C stocks in wetlands is less clear."

Comment: Ln 18: "growth due to low DO (Day. . .)"

*Authors' response:* We altered the sentence as suggested.

*Authors' changes:* Ln18-19: "Water saturation directly inhibits root growth due to low dissolved oxygen concentrations (Day and Megonigal, 1993; Tokarz and Urban, 2015)."

Comment: Ln 25: "distribution of high surface area minerals that are excellent sorbents for C in soils"

*Authors' response:* We thank the reviewer for this constructive edit and revised the sentence as suggested.

*Authors' changes:* Ln24-25: "In addition to restricting microbial metabolism and root growth, water saturation also influences the concentration and distribution of high surface area minerals that are excellent sorbents for C in soils …"

Pg 4

Comment: I point out three of our recent papers that are highly relevant to this introduction/discussion, but which were not likely available when this was drafted.

*Authors' response:* We thank the reviewer for pointing out these highly-relevant, recently published papers. We incorporated the following citations accordingly.

Comment: Ln 10: See Chen et al 2018 ES&T and Chen and Thompson 2018 ES&T on these topics

*Authors' response:* We incorporated these citations as requested.

*Authors' changes:* "Further, Al oxides, rather than Fe oxides, are the predominate mineral phases contributing to OM retention in forested floodplain sediments because their solubility is controlled by pH rather than redox conditions (Darke and Walbridge, 2000), and may thus play a critical role in mineral protection in seasonally flooded soils (Chen et al., 2018; Chen and Thompson, 2018)."

Comment: Ln 17: See Barcellos et al 2018 Soil Systems on this topic

*Authors' response:* We incorporated these citations as requested.

*Authors' changes:* "How the relationships between C and important biogeochemical controls differ in systems that undergo longer, yet not permanent, periods of water saturation is still in question – especially with depth (Barcellos et al., 2018)."

Comment: Ln 25: "measurements of soil. . ."

Authors' response: *We revised this sentence as suggested.*

Authors' changes: *Ln23-25: "To accomplish our first objective, we related soil $CO_2$ efflux at three landscape positions (upland, transition, and lowland) spanning the transect over the course of a full drainage and flooding cycle to measurements of soil temperature, moisture, water table depth and redox potential."*

Methods

Comment: Well done, except that more description of the stats used are required potentially to clarify issues I raise above and below with regard to Table 3 lowercase lettering.

Authors' response: *We agree that a more detailed description of the statistics used to generate table 3 are needed. In the revised version, we detail comprehensively how the data was binned and compared for this analysis. Briefly, included the R packages we used to determine letter designations using Tukey HSD comparisons (agricolae and MulticompView) where the alpha value was set at 0.05 for different letter designations. Within the analysis, the samples were tested for significance based on the landscape position and horizon. Additionally, as mentioned previously, we will include Table S7 in the supplemental material that provides the horizon-based ANOVA analysis mentioned above.*

Authors' changes: *' Statistical analyses were conducted on square root transformed data when assumptions of normal distribution were not met. Tukey's honestly significance difference (HSD) tests were conducted in Rstudio using packages agricolae (de Mendiburu, 2017), and multcompView (Graves et al., 2015) with an alpha value of 0.05 used for different letter designations. Due to the conservation nature of using a Tukey's HSD on our dataset, we additionally tested differences among horizon groups (i.e. across A or C/Cg horizons) using analysis of variance (ANOVA) tests (Table S7).'*

Results

Comment: Main issue in this section is the depth that is considered 'subsurface'. How does one determine this for soils with different depths or thicknesses? Normally, this doesn't matter, but in this case the authors are making a key argument about the C and Fe interactions and chemistry "at depth". Examining the C horizons, total C is actually higher in the lowland than in the upland and this would be true even if we examined C concentrations at 25 cm across the sites. If we go deeper, then the Cg of the transition and the lowland are equally low and the upland is higher, but not statistically higher based on the lower case letter assignments. The same is true in inverse for the lowest depth for Fe-o, it is highest at the lowland, but this is not significant from the other sites. This makes statements like "C concentrations were significantly lower in the lowland than in the upland subsoils", which is in the abstract, incorrect based on the authors' assignment of letters (see Table 3).

Authors' response: *We hope that additional explanation regarding the statistical tests and inferences provided above make it clear that there are statistical differences among A and C/Cg*

*horizons, respectively. This statement was not made based on the ANOVA results for the full data table, but for individual comparisons on a horizon-basis.*

Pg. 11

Comment: Ln 9: Assuming that Feb – June is the wet period, but you should tell readers that explicitly.

*Authors' response: The authors realize there had not been a clear definition of the growing and non-growing season designations within the methods section. We have added this definition to page 10 under statistical analyses. We added reminders in brackets on page 11.*

*Authors' changes: Pg10, Ln9-10: "Regression analyses were conducted for the entire year-long dataset, and for the growing and non-growing seasons defined as May through September and October through March, respectively."*

*Page 11:*
Comment: Ln8-9: Assuming that Feb – June is the wet period, but you should tell readers that explicitly.

Authors' response: We added a line to more clearly point out the flooded period of the lowland position in relation to the growing season parameter.

*Authors' changes: Ln9-10: "The flooded period (February through June) of the lowland position extended into the first two months of the growing season."*

Comment: Ln 11: "significantly lower than in. . ."

*Authors' response: We have revised the sentence, but where the reviewer suggested 'lower' it should actually remain 'greater'.*

*Authors' changes: Ln11-13:" This general difference became even more pronounced when cumulative $CO_2$ emissions were normalized to C content, with the upland position showing significantly greater emissions than in the transition (p-value <0.001; Tukey's HSD) and lowland (p-value <0.001, Tukey's HSD) positions."*

Comment: Ln 13: "season lowland VMC. . ."

*Authors' response: We revised the sentence as suggested.*

*Authors' changes: Ln21-22: "Soil moisture was consistently the greatest in the lowland position; during the growing season lowland VMC was 20% greater than the upland position (p-value < 0.05; Tukey's HSD), and 15% greater in the non-growing season (p-value < 0.05; Tukey's HSD) (Table S1)."*

Comment: Ln 21: Maybe it would be helpful to calculate the EH7 values here so that these could be compared with other studies and compared between the surface and subsurface horizons.

*Authors' response: The variations in pH values across our site is minimal and within the margin of error (4.98-5.43). Correcting the Eh values for pH would change Eh values by less than a decimal point. For example, an Eh value of 400 mV would correspond to Eh7 values of 399.91 mV at pH 5.43 and 399.88 mV at pH 4.98, respectively. These two pH values cover the range of values observed at our site, so we concluded that normalizing Eh to pH wouldn't change the numbers sufficiently to warrant inclusion.*

Comment: Ln 23: change "mineralogy" which is the study of minerals to "mineral composition". Do this elsewhere as well.

*Authors' response: We agree with this correction and have corrected it throughout the manuscript.*

Pg. 12

Comment: Ln 1: The data are more complex than this statement suggests. Please revise.

*Authors' response: We have revised the topic statement as suggested.*

*Authors' changes: Ln7: "3.3 Relating carbon concentration to root biomass and mineral composition across upland-to-lowland transitions"*

Comment: Ln 2: Do you mean topsoil here???? Because actually it is over 8 times the subsoil, but according to the letters, the lowland and transition topsoil are equal within error.

*Authors' response: We're thankful the reviewer caught this misspelling. We replaced subsoil with topsoil.*

*Authors' changes: Ln11-12: "C concentrations in the lowland position topsoil were two and four times greater than the transition (p-value < 0.01; Tukey's HSD) and upland positions topsoil (p-value < 0.001; Tukey's HSD), respectively."*

Comment: Ln 4: although this was not statistically significant, correct? I suggest adding that information.

Comment: Ln 7: Although again this was not statistically significant, right? Tell the reader that.

*Authors' response: In regards to the two comments above, we included a more detailed explanation of our statistical approaches as outlined in our response to the more general comments above. Again, these statements were supported by our horizon-based ANOVA results (Table S7). Using a Tukey's HSD (which is reported by letter designations in Table 3) is an overly conservative approach for our purposes as it accounts for the variance within the whole dataset. Conducting a horizon based ANOVA test allowed us to compare only the comparable horizons and include the variances within those horizons when testing for differences. The ANOVA results show a significant difference among positions when comparing the A-horizons*

*(p-value = 0.02), in the upland B- and transition and lowland C-horizons (p-value = 0.04) and the upland C- and transition and lowland Cg-horizons (p-value = 0.01). These findings support our statistical significance of the differences in C concentrations among the three positions in the adjacent horizons. To clarify the statistics for the reader, we have adjusted Table 3 to reflect the ANOVA results reported in Table S7. We have also added these p-values with the text ANOVA, to signify where the significance is derived. We have also included references to Table S7 when referring to the C concentration significance.*

*Authors' changes: Ln9-11: "Along the upland to lowland transects, C concentrations in the surface horizons significantly increased (p-value < 0.05, ANOVA) whereas concentrations in the subsurface horizons decreased along the transect (p-value < 0.05, ANOVA) (Table 3 and S7)."*

*Authors' changes: Ln12-15: "In contrast, the subsoils in the upland positions had nearly double the C concentrations than the subsoils of the transition and lowland positions (p-value < 0.05, ANOVA, Table S7)."*

Comment: Ln 9: True, except in the upland, right (Table 3 indicates it is not significant).

*Authors' response: Although we're not entirely sure, we believe the reviewer is referring to the silt and clay data. The statement has been adjusted in regards to significance.*

*Authors' changes: Ln14-16: "Silt and clay content increased from the upland to the lowland positions, particularly in the subsoil, although shifts in silt and clay contents were not statistically significant (+33%, Table 3)."*

Comment: Ln 12: Change 'determine' to 'predict'

*Authors' response: We have made the suggested change.*

*Authors' changes: Ln24: "To predict the relative influence of roots, mineral composition and $E_h$ on C concentrations in each landscape position…"*

Comment: Ln 18: "concentrations decreased along..."

*Authors' response: We have made the suggested change.*

*Authors' changes: Ln4-6: "The model results show that as the importance of redox-active $Fe_o$ as a predictor for soil C concentrations decreased along upland-to-lowland transects, the importance of $Al_o$ increased."*

Comment: Ln 20: Change 'identify' to 'predict'

*Authors' response: We have made the suggested change.*

*Authors' changes: Page13, Ln7: "To predict the influence of the biogeochemical variables on soil C concentrations with soil depth…"*

Comment: Ln 24: Maybe not Eh, but likely O2, right?

*Authors' response: We agree that it is oxygen availability, and not Eh, that is likely what causes higher C concentrations. We have adjusted the statement to reflect this detail.*

*Authors' changes: Page 13, Ln11-13: "These results indicate that, among the tested biogeochemical variables, $E_h$, a proxy for oxygen availability, has a predominant influence on C concentrations in the surface soils, while $Al_o$ has the strongest influence on C concentrations at depth. "*

Comment: Ln 25: Change 'effect' to 'influence'

*Authors' response: We have made the suggested change.*

*Authors' changes: Page 13, Ln11-12: "while $Al_o$ has the strongest influence on C concentrations*

*at depth."*

Pg. 13

Comment: Ln 17: "across the upland to lowland transect. . ."

*Authors' response:  We have made the suggested change.*

*Authors' changes: Page14, Ln4: "increases across the upland to lowland transect"*

Comment: Ln 18: ". . .(-11%) moving from upland to lowland."

*Authors' response: We have made the suggested change.*

*Authors' changes: Page 14, Ln5: "lipids decreases (-11%) moving from upland to lowland (Fig. 6b, Table S5)."*

Comment: Fig. 2: Symbols are hard to tell from one another. Consider using squares, triangles and circles. Cool could help too since other figures are in color.

*Authors' response: We thank the reviewer for this suggestion and have modified the figure to include color for a clearer designation between positions and depths.*

Comment: Fig. 2: Are the Eh values on these graphs corrected for pH? To allow comparisons between the depths/sites?

*Authors' response: We refer to our comment above about the minimal effect of pH on Eh values.*

Discussion

Pg. 13

Comment: Ln 22: Change 'demonstrate' to 'suggest'

Authors' response: We have made the suggested change.

Authors' changes: Page14, Ln10-11: "Our results suggest that the factors regulating $CO_2$ emissions and C accumulation shift…"

Comment: Ln 23: "...transects, but exhibit potentially inverse trends in the subsurface."

Authors' response: We have made the suggested change.

Authors' changes: Page 14, Ln10-12: "Our results suggest that the factors regulating $CO_2$ emissions and C accumulation shift as predicted in surface soils along the upland to lowland transects, but exhibit potentially inverse trends in the subsurface."

Pg. 14

Comment: Ln 1-2: delete sentence.

Authors' response: We deleted the suggested sentence.

Authors' changes: Page 15, deleted sentence "Our results show how seasonal flooding affects redox conditions, root biomass, and mineral composition as well as their impact on $CO_2$ efflux, C accumulation, and C chemistry across the upland to lowland transects."

Comment: Ln 4: "Our field data support our hypothesis that reducing. . ."

Authors' response: We have made the suggested change.

Comment: Ln 16-20: Clarify this section.

Authors' response: We revised this section for clarify.

Authors' changes: Page 15, Ln4-9: "Although the lowland soils become oxygenated in the non-growing season due to the water table drop, we observed near equal $CO_2$ emissions from the three landscape positions (Table 1, Fig. 2a). A possible explanation for this convergence in $CO_2$ emissions during the non-growing season could be the low seasonal temperatures (-1.7 to 10°C) which inhibit microbial activity. In other words, even when drainage allows for aerobic metabolism in the lowland soils, respiration rates still remain limited due to low temperatures."

Comment: Ln 24: Note that the figure shows topography that is not flat.

*Authors' response:* We appreciate this correction, and have updated the statement to reflect figure 1.

*Authors' changes: Page 15, Ln9-10: "Given the proximity of our three positions and minor change in elevation, aboveground litter inputs can be considered equal across the transect."*

Pg. 15

Comment: Ln 7: Consider using C stocks instead of concentration, which would help get around the depth issue.

*Authors' response: The authors agree that using C stocks would be beneficial. Due to the lack of accurate bulk density data for this site, however, we have chosen to focus on concentrations.*

Pg. 17

Comment: Ln 23: OK, but Cg in the lowland is 2nd highest across ALL sites/depths, so this statement doesn't ring fully true for me.

*Authors' response: Despite the Cg horizon in the lowland positions being the 2nd highest horizon, it still had overall lower concentrations of both extractable Fe and Al compared to the other positions. Which has led us to conclude that even though new C inputs could occur belowground, there is still a limited mineral protection capacity within the lowland positions.*

*Authors' changes:*

*Pg. 16, Ln 9-11: "An observed, but insignificant, increase in $Fe_o$ concentrations in the lowland $C_g$-horizons are likely a reflection of vertical transportation of soluble Fe phases. Despite this trend, there were overall lower concentrations of potentially reactive Fe and Al in the lowland $C_g$ horizon when taking into account both extractable Fe and Al contents (Table 3)."*

*Pg. 18, Ln 14-20: "Additionally, seasonal flooding over pedologic time scales has resulted in a total overall loss of reactive minerals and metals and thus diminished the potential capacity of the soils to accumulate C through other means. Recent studies suggest that colonization by deep-rooting upland plants will offset some of the C loss upon drainage of former wetlands through additional C inputs (Gorham et al., 1991; Lal 2008). The overall lower concentrations of reactive Fe and Al phases observed in seasonally flooded soils investigated here suggests a low capacity for new C inputs to associate with reactive Fe or Al phases, and, consequently, a low potential to offset the losses of anaerobically protected C upon drainage."*

Conclusions

Comment: Ln 10: change 'related to' to 'correlated with'

*Authors' response: We have made the suggested change.*

*Authors' changes:* Ln21-22: *"In the subsoil of seasonally flooded soils, anaerobic protection of C appears to be less important. C accumulation was low and primarily correlated with to $Al_o$,..."*

Comment: Ln 12: But, again what about Fe-o in the lowland Cg?????

*Authors' response:* *The reviewer points out here a trend in our dataset that we do not fully address, an apparent increase in $Fe_o$ within the Cg horizon in the lowland position. While it is clear that Fe is lower in the A and C horizons, this increase in Fe in the Cg horizon is likely due to vertical transport of soluble Fe. To address the slight increase in Feo in the lowland Cg we have added a comment in the 4.2 discussion section.*

*Authors' changes:* *Section 4.2, Page 16: Ln 9-10: "An observed, but insignificant, increase in $Fe_o$ concentrations in the lowland Cg-horizons are likely a reflection of vertical transportation of soluble Fe phases and reprecipitation at depth."*

---

## Author Response (AR1)

**RESPONSE TO EDITORIAL COMMENTS**

Dear Dr. Luo Yu,

Thank you for the opportunity to submit a revised manuscript to *Biogeosciences*. We have addressed the issues and concerns offered in in the two very thoughtful and constructive reviews as detailed in our point-by-point response below. A marked-up version of the revised manuscript follows our response below. Reviewer #1 had noted the discrepancy between the statistical results presented in the text and those reported in Table 3. That issue was simply due to the fact that we had mistakenly reported the results of statistical tests (ANOVA + Tukey's HSD) we had conducted on the entire dataset, rather than on a horizon-basis. We relied exclusively on this horizon-based comparison to assess our data for this manuscript. This problem was remedied by correcting the table to reflect the results of the horizon-based comparison and explaining the statistical approach in more detail. Reviewer #2 further requested substantial revisions of the abstract, which we rewrote almost completely. Reviewer #2 noted that the discussion section was repetitive and didn't sufficiently highlight the novelty of the results. In the revised manuscript, all discussion sections were significantly improved to sharpen the main points we would like to communicate. In addition, we completely rewrote the broader implications section to link our observations to potential climate change impacts on seasonally flooded mineral soils.

We hope that these changes adequately address your and the reviewers' concerns. Thank you for considering our revised manuscript for publication in *Biogeosciences*.

Sincerely,
Marco Keiluweit
Assistant Professor
University of Massachusetts Amherst

**RESPONSE TO REVIEWERS' COMMENTS**

Regular font:   original comment by the reviewer
*Italized text:*           response by the authors
*"Italicized quotes":*   revised text segments

**Response to Comments by Referee #1**

*Comment:* Overall, I think this is a valuable dataset and a well-executed study. I support its eventual publication. The surface horizon data are well established, and I have just minor comments there as indicated below. However, the subsurface depth data are problematic, mostly because there are different overall soil depths in each of the sites and different horizon designations. This has led also I think to some statements that are not well supported by the statistics or that the statistics used are not well presented. For instance, reading the abstract while looking at Table 3 raises several questions if we interpret "significantly lower" to mean different statistical lowercase letter assignments, which most readers will. Much of my confusion occurs in section 3.3, where it appears in most cases the differences described are not statistically significant as shown on Table 3, but this is not pointed out in the text. At one point here the authors refer to a Tukey test for the topsoils (although I think they misplaced the word subsoil on pg. 12, ln 2) with a p value < 0.01 for the transition vs. lowland, but the those share a lowercase letter assignment in Table 3, which suggests they would not have a p value < 0.01. Correcting these presentation or interpretation issues is critical.

*Authors' response: We thank Dr. Thompson for these extremely thoughtful and constructive comments. We took the following steps to assure that the statistical tests and the resulting estimates of significance are reported correctly.*

*First, we emphasize that the statistical analyses were done on square root transformed data, but the data provided in Table 3 is the non-transformed observational data. We have clarified this in the methods section on Page 10: "Statistical analyses were conducted on square root transformed data when assumptions of normal distribution were not met, although non-transformed observational values are reported within the text (Table 3)."*

*Second, we clarified our statements regarding the significant differences in C contents among surface and subsurface soils across the upland-to-lowland transect. We believe that the confusion arose from the fact that we made statements in the text (both abstract and main body) that were not reflected in the statistics reported in Table 3. We had initially conducted ANOVAs and Tukey's HSD tests on the whole dataset (i.e., across all landscape positions and horizons). But as we were primarily concerned with comparing C contents across individual horizons along the transect (e.g., comparing A horizons across upland, transition and lowland position), we conducted our final ANOVAs on a horizon-basis. Those tests indicated that both A and C horizons in upland and lowland positions are significantly different. We had presented the results from this analysis, including the p values indicating significant differences among the horizons, in the abstract (Ln: 20-22) and results section (Pg. 11 Ln:21-25). Unfortunately, the initial Table 3 mistakenly showed the results of the ANOVAs for the entire dataset, rather than the horizon-*

*based comparisons we used to test our hypotheses. The same mistake was made in Table S2 and S4. In the revised version, we updated the table letter designations, added detail on the statistical method, indicated which comparisons were made in the result section, and revised table footnotes to indicate that letter designations reflect the results from the horizon-basis analysis (Table 3, Table S2 and Table S4).*
5 *We hope that this explanation and the measures taken will clarify this issue.*

Assuming the stats letter values are correct, I think this could be resolved by looking at C stocks rather than C concentrations at depth and backing off on some of the subsurface interpretations that are not fully supported by the stats.

*Authors' response: We thank the reviewer for this suggestion. We believe the above explanation of the*
10 *statistical approach taken to support our interpretations resolves the confusion about the stats letters.*

If the authors have data binned at finer depth intervals, that might help clarify things as well, but if not I suggest using C stock down to 68 cm, in which case one could compare equally across all the sites. One could examine surface C stock (0- 25 cm) and then a subsurface C stock value (25 – 68cm). Outside of this major issue, I think the paper has a lot of promise and the combination of field CO2 data and
15 molecular-scale carbon chemistry is exciting.

*Authors' response: We thank the reviewer for this suggestion, but believe that the measures taken above to clarify our statistical approach sufficiently address this concern.*

Abstract

Comment: I read and reviewed the abstract without looking at any other parts of the MS to mimic a reader
20 looking at the abstract on-line. Read alone, I am not clear on the findings and implications and thus the abstract needs to be clarified. I give a couple of specifics below in the line edits, but I encourage the authors to have someone unfamiliar with the study read the abstract alone after revision.

*Authors' response: We revised the abstract in response to the specific comments provided below, but we have also substantially revised the abstract in response to the general comments provided. Most*
25 *importantly, added specific results that allow the reader to assess how we arrived at our conclusions.*

Comment: Ln 14: Is it really true that this is largely unknown? If this is just for seasonally flooded mineral soils (compared to wetlands in general), then this point escaped me on the first read. Perhaps it was the shift from "seasonally flooded soils" in the previous sentence to "seasonally flooded mineral soils" in this
30 sentence. Use one term and stick with in, especially in the abstract where space is tight.

*Authors' response: We agree that the terminology should be more consistent and are using  'seasonally flooded mineral soils' here and throughout the revised manuscript.*

*Authors' changes: Page 1, Ln12-14 "Redox conditions, plant root dynamics, and the abundance of protective mineral phases are well-established controls on soil C persistence, but their relative influence in seasonally flooded mineral soils is largely unknown."*

Comment: Ln 16: Need to specify here that the lowlands are periodically flooded and the uplands are not if that is indeed the case. I am assuming that, but one could have uplands that are also periodically flooded due to high rainfall and perched water tables.

*Authors' response: This point has been clarified in the abstract. The lowlands are in fact seasonally flooded whereas the transition and upland position are not.*

*Authors' changes: Page 1, Ln16-18 "Specifically, we contrasted mineral soils under temperate deciduous forests in lowland positions that undergo seasonal flooding with adjacent upland soils that do not, considering both surface (A) and subsurface (B/C) horizons."*

Comment: Ln 17: This sentence is hard to follow. I read it twice and was still not sure what it was saying, where C was higher? I suggest "We found the lowlands had lower CO2 effluxes than the uplands. Lowland surface soils (0-20 cm...or whatever it is, also could give A or B or O classification) had higher C concentrations a higher abundance. . .than the uplands."

*Authors' response: We revised the sentence as suggested.*

*Authors' changes: Page 1, Ln18-21 "We found the lowland soils had lower total annual $CO_2$ efflux than the upland soils, with monthly $CO_2$ efflux most strongly correlated with redox potential ($E_h$). Lower $CO_2$ efflux as compared to the uplands corresponded to greater C concentration and abundance of lignin-rich, higher-molecular weight, chemically-reduced organic compounds in the lowland surface soils (A-horizons)."*

Comment: Ln 20: Here I was confused again by subsoils slipping in there. I think you need to be much more upfront about this distinction as it is one of the main points of the abstract. At the end you also start to talk about C stocks (depth integrated concentrations), which would take into account bulk density. Consider discussing that here instead of concentration?

*Authors' response: We revised the abstract to more clearly distinguish subsurface from surface horizon results as well as C stocks from C concentration.*

*Authors' changes: Page 1, Ln25-30 "Combined, our results suggest that low redox potentials are the primary cause for C accumulation in seasonally flooded surface soils, likely due to selective preservation of organic compounds under anaerobic conditions. In seasonally flooded subsurface soils, however, C accumulation is limited due to lower C inputs through root biomass and the removal of Fe phases under reducing conditions."*

Comment: Ln23: It is not clear what non-reducible Al phases are being relied on for here? I assume mineral protection, but best not to have readers assuming in the abstract.

*Authors' response: The abstract was completely rewritten during revisions. We know specify that $Fe_o$ and $Al_o$ serve as proxies for protective metal phases. We further added a statement to clarify why we conclude that $Al_o$ becomes more important in lowland soils.*

*Authors' changes: Page 1, Ln23-25 "Our linear mixed effects model showed that $Fe_o$ served as the strongest measured predictor of C concentrations in upland soils, yet $Fe_o$ had no predictive power in lowland soils. Instead, our model showed that $E_h$ and $Al_o$ became significantly stronger predictors in the lowland soils."*

Comment: Ln 24-25: The three reasons given for why you see more C in the topsoils than the subsoils are not supported in the abstract by any data. Either include this data upfront (i.e., lowland had low/zero O2, whereas uplands had O2 above X%; also data on roots and Fe presence/abundance) or you could simply state that these C findings correlated with O2, roots and Fe, implying the data is in the paper, but not fully presented. What you are asking the reader to do here is accept this statement without any sense that it is supported by data in this paper and that is not comfortable to many readers (and me I suppose).

*Authors' response: We recognize that the statements within the abstract are not presented with the data to support these claims. We have revised the abstract to reflect that there were correlations found between C and roots, Fe and redox potential and the supporting data is found in the main manuscript.*

*Authors' changes: Page 1, Ln20-25 "Lower $CO_2$ efflux as compared to the uplands corresponded to greater C concentration and abundance of lignin-rich, higher-molecular weight, chemically-reduced organic compounds in the lowland surface soils (A-horizons). In contrast, subsurface soils in the lowland position ($C_g$-horizons) showed lower C concentrations than the upland positions, coinciding with lower abundance of root biomass and oxalate-extractable Fe phases ($Fe_o$, a proxy for reactive Fe). Our linear mixed effects model showed that $Fe_o$ served as the strongest measured predictor of C concentrations in upland soils, yet $Fe_o$ had no predictive power in lowland soils. Instead, our model showed that $E_h$ and $Al_o$ became significantly stronger predictors in the lowland soils"*

Comment: Ln26: Again, without O2 data or mineral protection data, how could you conclude this. I assume it is in the rest of the paper. . .but I have not read that yet if I am most readers.

*Authors' response: We have addressed this issue in our revision of the abstract.*

Introduction

Comment: The introduction does a nice job setting the stage although I suggest line edits below.

*Authors' response: We appreciate this comment and the subsequent revision suggestions for clarity.*

Pg. 2

Comment: Ln 14-15: Revise for clarity.

*Authors' response: The authors have revised the sentence as suggested.*

*Authors' changes: Page 3, Ln13-14 "Seasonal wetlands can be considered early warning ecosystems (Brooks, 2005); forecasting the impacts of climate change on permanently flooded mineral wetlands."*

Comment: Ln 16: Maybe not "model ecosystems" but essential "endmembers".

*Authors' response: We agree that 'endmembers' is a more suitable representation of these ecosystems.*

*Authors' changes: Page 3, Ln14-16 "Thus, seasonally flooded wetlands represent essential endmembers to study the effects of climate change on larger permanently flooded wetland soils (Brooks, 2005)."*

Comment: Ln 19: This is an "endash" and you want an "emdash" here. A longer dash, that should not have spaces around it. On MS-Word you hit dash twice between words without adding spaces and word turns it into an emdash. Do this elsewhere in the text.

*Authors' response: Thank you for noticing and correcting the endash/emdash inaccuracies. We have corrected all throughout the text.*

Comment: Ln24: "seasonal wetlands" or seasonally flooded mineral wetlands, choose one term and stick with it through-out the MS.

*Authors' response: We agree the terminology needs to be more consistent. We have revised the manuscript to be more consistent with the term 'seasonally flooded mineral soils'*

*Authors' changes: Page 3, Ln23-24 "Determining the controls on C cycling within seasonally flooded mineral soils thus requires specific consideration of the fluxes and dynamics across these terrestrial-aquatic transitions."*

Pg. 3

Comment: Ln 5: "catalyze"

*Authors' response: We revised the sentence, replacing "catalyzing.." to "..which catalyze..".*

*Authors' changes: Page 4, Ln4-6 "The resulting oxygen limitations inhibit the activity of oxidative enzymes which catalyze the depolymerization of higher-molecular weight OM into smaller, assimilable compounds (Megonigal et al. 2003)."*

Comment: Ln 10: instead of chemically-reduced, "lower valance" would be more precise.

*Authors' response: The authors appreciation this suggestion. To be consistent with other publications on the subject, we would like to retain the 'chemically-reduced' terminology, but also reference "oxidation state" in the revised version.*

*Authors' changes: Page 4, Ln7-9 "Anaerobic conditions limit microbes to utilizing substrates that are chemically more oxidized, in turn preferentially preserving more chemically-reduced organic compounds (i.e., compounds with lower C oxidation states) in soils and sediments (Boye et al. 2017; Keiluweit et al., 2017)."*

Comment: Ln 17: ", but the impact of roots on soil C. . ."

*Authors' response: We thank the referee for this revision, which adds more clarity to the sentence.*

*Authors' changes: Page 4, Ln16-17 "Roots are the main contributors to C stocks in upland soils (Rasse et al., 2005), but the impacts of roots on soil C stocks in wetlands is less clear."*

Comment: Ln 18: "growth due to low DO (Day. . .)"

*Authors' response: We altered the sentence as suggested.*

*Authors' changes: Page 4, Ln17-18 "Water saturation directly inhibits root growth due to low dissolved oxygen concentrations (Day and Megonigal, 1993; Tokarz and Urban, 2015)."*

Comment: Ln 25: "distribution of high surface area minerals that are excellent sorbents for C in soils"

*Authors' response: We thank the reviewer for this constructive edit and revised the sentence as suggested.*

*Authors' changes: Page 4, Ln23-24 "In addition to restricting microbial metabolism and root growth, water saturation also influences the concentration and distribution of high surface area minerals that are excellent sorbents for C in soils …"*

Pg 4

Comment: I point out three of our recent papers that are highly relevant to this introduction/discussion, but which were not likely available when this was drafted.

*Authors' response: We thank the reviewer for pointing out these highly-relevant, recently published papers. We incorporated the following citations accordingly.*

Comment: Ln 10: See Chen et al 2018 ES&T and Chen and Thompson 2018 ES&T on these topics

*Authors' response: We incorporated these citations as requested.*

*Authors' changes:* Page 5, Ln8-11 *"Further, Al rather than Fe oxides are the predominate mineral phases contributing to OM retention in forested floodplain sediments because their solubility is controlled by pH rather than redox conditions (Darke and Walbridge, 2000), and may thus play a critical role in mineral protection in seasonally flooded soils (Chen et al., 2018; Chen and Thompson, 2018)."*

Comment: Ln 17: See Barcellos et al 2018 Soil Systems on this topic

*Authors' response: We incorporated these citations as requested.*

*Authors' changes: Page 5, Ln15-16 "How the relationships between C and important biogeochemical controls differ in systems that undergo longer, yet not permanent, periods of water saturation is still in question – especially with depth (Barcellos et al., 2018)."*

Comment: Ln 25: "measurements of soil. . ."

*Authors' response: We revised this sentence as suggested.*

*Authors' changes: Page 5, Ln21-23 "To accomplish our first objective, we related soil $CO_2$ efflux at three landscape positions (upland, transition, and lowland) spanning the transect over the course of a full drainage and flooding cycle to measurements of soil temperature, moisture, water table depth and redox potential."*

Methods

Comment: Well done, except that more description of the stats used are required potentially to clarify issues I raise above and below with regard to Table 3 lowercase lettering.

*Authors' response: As indicated above, we mistakenly had reported the ANOVA results for the whole dataset in Table 3, even though our results/discussion in the text were all based on horizon-based ANOVAs. We have updated the lettering and footnotes in Table 3. We further detail our horizon based approach in the revised methods section.*

*Authors' changes: Page 11, Ln8-15 "To test our hypotheses, differences among landscape positions were assessed individually for each set of horizons using analyses of variance (ANOVA) conducted in Rstudio (version 5.3.1) combined with Tukey's honesty significance difference (HSD) tests using R packages agricolae (de Mendiburu, 2017) and multcompView (Graves et al., 2015). Specifically, we compared values within surface (A), intermediate (B/C), and subsurface (C/$C_g$) horizons across the upland-to-lowland transect (Table 3). Alpha values of 0.05 were used for letter designations indicating significant differences among the landscape positions. Statistical analyses were conducted on square-root transformed data when assumptions of normal distribution were not met."*

Results

Comment: Main issue in this section is the depth that is considered 'subsurface'. How does one determine this for soils with different depths or thicknesses? Normally, this doesn't matter, but in this case the authors are making a key argument about the C and Fe interactions and chemistry "at depth". Examining the C horizons, total C is actually higher in the lowland than in the upland and this would be true even if we examined C concentrations at 25 cm across the sites. If we go deeper, then the Cg of the transition and the lowland are equally low and the upland is higher, but not statistically higher based on the lower case letter assignments. The same is true in inverse for the lowest depth for Fe-o, it is highest at the lowland, but this is not significant from the other sites. This makes statements like "C concentrations were significantly lower in the lowland than in the upland subsoils", which is in the abstract, incorrect based on the authors' assignment of letters (see Table 3).

*Authors' response: We hope that additional explanation regarding the statistical tests and inferences provided above make it clear that there are statistical differences among A and C/Cg horizons, respectively. This statement was not made based on the ANOVA results for the full dataset we initially reported in Table 3, but for individual comparisons on a horizon-basis. We have updated Table 3 and other data tables to accurately reflect the horizon-based comparison.*

Pg. 11

Comment: Ln 9: Assuming that Feb – June is the wet period, but you should tell readers that explicitly.

*Authors' response: The authors realize there had not been a clear definition of the growing and non-growing season designations within the methods section. We have added this definition to page 10 under statistical analyses. We added reminders in brackets on page 11.*

*Authors' changes: Page 10, Ln24-25 "Regression analyses were conducted for the entire year-long dataset, and for the growing and non-growing seasons defined as May through September and October through March, respectively."*

*Page 11:*
Comment: Ln8-9: Assuming that Feb – June is the wet period, but you should tell readers that explicitly.

*Authors' response: We added a line to more clearly point out the flooded period of the lowland position in relation to the growing season parameter.*

*Authors' changes: Page 11, Ln22-24 "The flooded period (February through June) of the lowland position extended into the first two months of the growing season."*

Comment: Ln 11: "significantly lower than in. . ."

*Authors' response: We have revised the sentence, but where the reviewer suggested 'lower' it should actually remain 'greater'.*

*Authors' changes: Page 11-12, Ln24-1"This general difference became even more pronounced when cumulative $CO_2$ emissions were normalized to C content, with the upland position showing significantly greater emissions than in the transition (p-value <0.001; Tukey's HSD) and lowland (p-value <0.001, Tukey's HSD) positions."*

Comment: Ln 13: "season lowland VMC. . ."

*Authors' response: We revised the sentence as suggested.*

*Authors' changes: Page 12, Ln11-13 "Soil moisture was consistently the greatest in the lowland position; during the growing season lowland VMC was 20% greater than the upland position (p-value < 0.05; Tukey's HSD), and 15% greater in the non-growing season (p-value < 0.05; Tukey's HSD) (Table S1)."*

Comment: Ln 21: Maybe it would be helpful to calculate the EH7 values here so that these could be compared with other studies and compared between the surface and subsurface horizons.

*Authors' response: The variations in pH values across our site is minimal and within the margin of error (4.98-5.43). Correcting the Eh values for pH would change Eh values by less than a decimal point. For example, an Eh value of 400 mV would correspond to Eh7 values of 399.91 mV at pH 5.43 and 399.88 mV at pH 4.98, respectively. These two pH values cover the range of values observed at our site, so we concluded that normalizing Eh to pH wouldn't change the numbers sufficiently to warrant inclusion.*

Comment: Ln 23: change "mineralogy" which is the study of minerals to "mineral composition". Do this elsewhere as well.

*Authors' response: We agree with this correction and have corrected it throughout the manuscript.*

Pg. 12

Comment: Ln 1: The data are more complex than this statement suggests. Please revise.

*Authors' response: We have revised the topic statement as suggested.*

*Authors' changes: Page 12, Ln7 "3.3 Relating carbon concentration to root biomass and mineral composition across upland-to-lowland transitions"*

Comment: Ln 2: Do you mean topsoil here???? Because actually it is over 8 times the subsoil, but according to the letters, the lowland and transition topsoil are equal within error.

*Authors' response: We're thankful the reviewer caught this misspelling. We revised accordingly.*

*Authors' changes: Page 12, Ln24-25 "C concentrations in the lowland position surface horizons were two and four times greater than the transition (p-value < 0.01; Tukey's HSD) and upland positions surface horizons (p-value < 0.001; Tukey's HSD), respectively."*

Comment: Ln 4: although this was not statistically significant, correct? I suggest adding that information.

Comment: Ln 7: Although again this was not statistically significant, right? Tell the reader that.

*Authors' response:  In regards to the two comments above, we included a more detailed explanation of our statistical approaches as outlined in our response to the more general comments above.  Again, these statements were supported by our horizon-based ANOVA results which are now shown in the revised Table 3. Conducting a horizon-based ANOVA test allowed us to compare only the comparable horizons and include the variances within those horizons when testing for differences. The Tukey's HSD results show a significant difference between the upland and lowland A-horizons (p-value = 0.03) and the upland C- and lowland Cg-horizons (p-value = 0.02). These findings support our statistical significance of the differences in C concentrations among the upland and lowland adjacent horizons. In addition to correcting Table 3, we revised Table S2 and S4 accordingly and made sure that the tables and results sections 3.3, 3.4 and 3.5 are consistent.*

*Authors' changes: Page 12, Ln22-24 "Along the upland to lowland transects, C concentrations in the surface horizons significantly increased (p-value < 0.05, Tukey's HSD) whereas concentrations in the subsurface horizons decreased along the transect (p-value < 0.05, Tukey's HSD) (Table 3)."*

*Authors' changes: Page13, Ln: 1-2 "In contrast, the subsurface soils in the upland positions had nearly double the C concentrations than in the transition and lowland positions (p-value < 0.05, Tukey's HSD) (Table 3)."*

Comment: Ln 9: True, except in the upland, right (Table 3 indicates it is not significant).

*Authors' response: Although we're not entirely sure, we believe the reviewer is referring to the silt and clay data. The statement has been adjusted in regards to significance.*

*Authors' changes: Page 13, Ln4-6 "Silt and clay content increased from the upland to the lowland positions, particularly in the subsoil, although shifts in silt and clay contents were not statistically significant (+33%, Table 3)."*

Comment: Ln 12: Change 'determine' to 'predict'

*Authors' response: We have made the suggested change.*

*Authors' changes: Page 13, Ln14 "To predict the relative influence of roots, mineral composition and $E_h$ on C concentrations in each landscape position…"*

Comment: Ln 18: "concentrations decreased along..."

*Authors' response: We have made the suggested change.*

*Authors' changes: Page 13, Ln19-21 "The model results show that as the importance of redox-active $Fe_o$ as a predictor for soil C concentrations decreased along upland-to-lowland transects, the importance of $Al_o$ increased."*

Comment: Ln 20: Change 'identify' to 'predict'

*Authors' response: We have made the suggested change.*

*Authors' changes: Page13, Ln22 "To predict the influence of the biogeochemical variables on soil C concentrations with soil depth…"*

Comment: Ln 24: Maybe not Eh, but likely O2, right?

*Authors' response: We agree that it is oxygen availability, and not Eh, that is likely what causes higher C concentrations. We have adjusted the statement to reflect this detail.*

*Authors' changes: Page 14, Ln1-2 "These results indicate that, among the tested biogeochemical variables, $E_h$, a proxy for oxygen availability, has a predominant influence on C concentrations in the surface soils, while $Al_o$ has the strongest influence on C concentrations at depth. "*

Comment: Ln 25: Change 'effect' to 'influence'

*Authors' response: We have made the suggested change.*

*Authors' changes: Page 13, Ln11-12 "while $Al_o$ has the strongest influence on C concentrations at depth."*

Pg. 13

Comment: Ln 17: "across the upland to lowland transect. . ."

*Authors' response: We have made the suggested change.*

*Authors' changes: Page14, Ln6 "increases across the upland to lowland transect"*

Comment: Ln 18: ". . .(-11%) moving from upland to lowland."

*Authors' response: We have made the suggested change.*

*Authors' changes: Page 14, Ln20-23 "Paralleling that change, the relative contributions of lignin increased (+7%) and that of lipids decreased (-11%) moving from upland to lowland position (Fig. 6b, Table S5)"*

Comment: Fig. 2: Symbols are hard to tell from one another. Consider using squares, triangles and circles. Cool could help too since other figures are in color.

*Authors' response: We thank the reviewer for this suggestion and have modified the figure to include color for a clearer designation between positions and depths.*

Comment: Fig. 2: Are the Eh values on these graphs corrected for pH? To allow comparisons between the depths/sites?

*Authors' response: We refer to our comment above about the minimal effect of pH on Eh values.*

Discussion

*Authors' response: Please note that in response to Reviewer #1, we have substantially revised sections 4.1, 4.2 and 4.3.*

Pg. 13

Comment: Ln 22: Change 'demonstrate' to 'suggest'

*Authors' response: We have made the suggested change.*

*Authors' changes: Page15, Ln4-5 "Our results suggest that the factors regulating $CO_2$ emissions and C accumulation shift…"*

Comment: Ln 23: "...transects, but exhibit potentially inverse trends in the subsurface."

*Authors' response: We have revised the statement leading into the discussion section..*

*Authors' changes: Page 15, Ln5 "Our results suggest that the factors regulating $CO_2$ emissions and C accumulation shifted as predicted in surface soils along the upland-to-lowland transects, but exhibited inverse trends in the subsurface."*

Pg. 14

Comment: Ln 1-2: delete sentence.

Authors' response: We deleted the suggested sentence.

Authors' changes: Page 15, deleted sentence "Our results show how seasonal flooding affects redox conditions, root biomass, and mineral composition as well as their impact on $CO_2$ efflux, C accumulation, and C chemistry across the upland to lowland transects."

Comment: Ln 4: "Our field data support our hypothesis that reducing. . ."

Authors' response: We have made the suggested change.

Author's changes: Page 15, Ln5-6 "Our field data support our hypothesis that reducing conditions under flooded conditions inhibit microbial respiration and thus reduce $CO_2$ emissions in the lowland position"

Comment: Ln 16-20: Clarify this section.

Authors' response: This section was comprehensively revised and significantly shortened for clarity.
Authors' changes: Page 15, Ln6-19 "Indeed, seasonal $CO_2$ emissions in the lowland positions were strongly correlated with VMC, water table depth, and $E_h$ (Fig. 3). Conversely, in upland positions where oxygen limitations are not limiting, soil temperature was found to the best predictor variable for $CO_2$ emissions (Fig. 3a, Table 2). Our results further indicated that the impact of seasonal drainage of the lowland soils on $CO_2$ effluxes is limited by temperature effects (Fig. 2, Table 2). Oxygenation in other seasonally flooded soils usually results in increases in $CO_2$ effluxes due to enhanced aerobic microbial respiration (Laine et al., 1996; Krauss and Whitbeck, 2012). Although our lowland soils become oxygenated in the non-growing season due to the water table drop, we observed near equal $CO_2$ emissions from the three landscape positions (Table 1, Fig. 2a). A likely explanation for this convergence in $CO_2$ emissions is that oxygenation coincides with the low seasonal temperatures during the non-growing season (-1.7 to 10 degrees Celsius) which inhibit microbial activity (Lloyd and Taylor, 1994). In other words, even when seasonal drainage oxygenates the lowland soils, potentially allowing for aerobic microbial respiration to occur, $CO_2$ efflux in these seasonal wetlands still remains suppressed due to cold temperatures. It remains to be seen if shifts in the drainage period or higher temperatures during the non-growing season, as expected throughout the Northeastern US with continued global warming, disproportionally increase microbial respiration (and potentially C loss) from these soils."

Comment: Ln 24: Note that the figure shows topography that is not flat.

Authors' response: We appreciate this correction, and have updated the statement to reflect figure 1.

Authors' changes: Page 15, Ln23-25 "Given the proximity of our three positions and minor change in elevation, aboveground litter inputs can be considered equal across the transect."

Pg. 15

Comment: Ln 7: Consider using C stocks instead of concentration, which would help get around the depth issue.

*Authors' response: The authors agree that using C stocks would be beneficial. Due to the lack of accurate bulk density data for this site, however, we have chosen to focus on concentrations.*

Pg. 17

Comment: Ln 23: OK, but Cg in the lowland is 2nd highest across ALL sites/depths, so this statement doesn't ring fully true for me.

*Authors' response: Despite the Cg horizon in the lowland positions being the 2nd highest horizon, it still had overall lower concentrations of both extractable Fe and Al compared to the other positions. Which has led us to conclude that even though new C inputs could occur belowground, there is still a limited mineral protection capacity within the lowland positions.*

*Authors' changes:*

*Page 16, Ln21-23 "A noticeable, yet insignificant, increase in $Fe_o$ concentrations in the lowland Cg-horizons (Table 3) is likely a reflection of vertical transport of soluble or colloidal Fe phases into the subsurface horizon, where they may reprecipitate during drained periods."*

*Pg. 18, Ln 13-15: "As noted above, while the lowland horizons overall showed a decline in $Fe_o$ and $Al_o$ contents relative to the upland position (Table), a modest uptick in $Fe_o$ content was observed in the lowland subsurface (Cg) horizon. One possibilities is thus that Fe precipitates in the seasonally flooded subsurface soil horizon trapped dissolved, partially-oxidized, lignin-derived OM leaching down the profile and so resulted in the accumulation of relatively oxidized OM"*

Conclusions

Comment: Ln 10: change 'related to' to 'correlated with'

*Authors' response: We have made the suggested change.*

*Authors' changes: Page 19, Ln16 "In the subsoil of seasonally flooded soils, anaerobic protection of C appears to be less important. C accumulation was low and primarily correlated with to $Al_o$,…"*

Comment: Ln 12: But, again what about Fe-o in the lowland Cg?????

*Authors' response: The reviewer points out here a trend in our dataset that we do not fully address, an apparent increase in $Fe_o$ within the Cg horizon in the lowland position. While it is clear that Fe is lower*

*in the A and C horizons, this increase in Fe in the Cg horizon is likely due to vertical transport of soluble Fe. To address the slight increase in Feo in the lowland Cg we have added a comment in the 4.2 discussion section.*

5    *Authors' changes: Page 16, Ln25-26, "An observed, but insignificant, increase in $Fe_o$ concentrations in the lowland Cg-horizons are likely a reflection of vertical transportation of soluble Fe phases and reprecipitation at depth."*

**Response to Comments by Referee #2**

Comment: LaCroix et al. report findings on C storage and changes in the physical-geochemical composition of soils minerals and redox conditions under the seasonal flooding soils. Nevertheless, in different part of the manuscript several strongly weak points have been identified that must be addressed
5 from the authors. Moreover, I do not find that results provided insight into the mechanism on C storage and the changes in the shifting minerals and other critical factors.

*Authors' response: We thank the reviewer for taking the time to review out manuscript and believe we adequately addressed the specific concerns mentioned below.*

Comment: 1. Synchrotron-based X-ray analyses and FT-ICR-MS analyses. There are plenty of literatures
10 on these methods, as a reviewer also a reader, I suggest simplify these parts and move the detail descriptions into SI.

*Authors' response: Biogeosciences allows for detailed methods sections. In the interest of providing sufficient information to less informed readers, we would like keep the level of detail in the methods section as is.*

15 Comment: 2. Results. In figure 2, the symbols are too small and similar to be recognized, while and the resolution are lower.

*Authors' response: We have updated figure 2 to include color to differentiate the different positions and depths, and to be more legible in general.*

Comment: 3. In 3.3, compared with Feo and Alo, the authors haven't presented the detailed data of Fed
20 and Ald, which are more sensitive to the changing of environmental factors. Meanwhile, the soil iron cycling is sensitive to the seasonal flooding, the recrystalliza- tion processes of iron oxides as well as aluminum oxides during the shifting of seasonal flooding soils are critical factors to the variation of the iron/aluminum species, which are further controlled the reactivity of iron/aluminum species in soil environment.

25 *Authors' response: As discussed in the initial manuscript (Page 12 Ln13-14), the dithionite-extractable Fe and Al data is well correlated with the amount of oxalate-extractable Fe. In other words, the trends are the same. We thus believe that adding the dithionite extraction data to the main manuscript has no added values and left them in the SI.*

Comment: 4. In 3.4, carboxylic/aromatic C ratios is a suitable indicator to present the different of
30 oxidation degrees, however, from fig 5b, it's inaccurate to describe the increase trends of those values above in C horizons. It doesn't present significant different in fig 5b.

*Authors' response: We changed the text to reflect that these differences are not significant at the $p < 0.05$ values, but maintain that the trends are ecologically relevant and worthy of discussion. We would also*

*like to note that this is one of the first manuscripts that reports variability in analyses such as C NEXAFS and FTICRMS. Due to the low throughput of these analyses, replicates are generally pooled before analysis and only analyzed as a composite sample in other publications, leaving the reader with no sense of the natural variability.*

*Authors' changes: Page 14, Ln8-12 "Although the ratios were not significantly different among the landscape positions ($p > 0.05$, Tukey's HSD), noteworthy trends were found. In the surface horizons, the ratio gradually decreased across the upland-to-lowland transects in the surface horizons (Fig. 5b, Table S4). In the subsurface horizons, the opposite trend was observed, and the ratio steadily increased from the upland C horizons to the lowland Cg horizons (Fig. 5b, Table S4)."*

Comment: 5. In discussion part, the authors just repeated the obtained results described in results part in another similar way, lacking of further discussion around the mechanism among C storage and the changes in the shifting minerals and other critical factors. As a reader, I find it's hard to get new information in this important parts.

*Authors' response: In response to this concern, we substantially revised the discussion sections to add information regarding the mechanisms under consideration. In particular, we substantially shortened section 4.1, comprehensively revised sections 4.2 and 4.3, and completely rewrote section 4.4.*

Comment: 6. Further experiments should be designed and conducted to illustrate the mechanism on soil chemical-physical properties and C storage.

*Authors' response: We believe that in-field measurements and analysis of field samples along environmental gradients is a suitable tool to examine the effects of different soil properties on soil carbon. While we acknowledge that this approach is based on correlations, and mechanisms can thus merely be inferred, it is one that is widely used in the biogeosciences community (Angst et al., 2018; Barcellos et al., 2018; Chen et al., 2017; Hall et al., 2016, Hall & Silver 2015; Olshansky et al., 2018; Torn et al. 1997). We believe this study uniquely combines environmental data with soil physical-chemical properties to infer variations in C storage mechanisms across upland-to-lowland transects.*

*Full references:*

Angst, G., Messinger, J., Greiner, M., Häusler, W., Hertel, D., Kirfel, K., Kögel-Knabner, I., Leuschner, C., Rethemeyer, J. and Mueller, C. W.: Soil organic carbon stocks in topsoil and subsoil controlled by parent material, carbon input in the rhizosphere, and microbial-derived compounds, Soil Biology and Biochemistry, 122, 19–30, doi:10.1016/j.soilbio.2018.03.026, 2018.

Barcellos, D., O'Connell, C., Silver, W., Meile, C. and Thompson, A.: Hot Spots and Hot Moments of Soil Moisture Explain Fluctuations in Iron and Carbon Cycling in a Humid Tropical Forest Soil, Soil Systems, 2(4), 59, doi:10.3390/soilsystems2040059, 2018.

Chen, C., Kukkadapu, R. K., Lazareva, O. and Sparks, D. L.: Solid-Phase Fe Speciation along the Vertical Redox Gradients in Floodplains using XAS and Mössbauer Spectroscopies, Environmental Science & Technology, 51(14), 7903–7912, doi:10.1021/acs.est.7b00700, 2017.

5   Hall, S. J. and Silver, W. L.: Reducing conditions, reactive metals, and their interactions can explain spatial patterns of surface soil carbon in a humid tropical forest, Biogeochemistry, 125(2), 149–165, doi:10.1007/s10533-015-0120-5, 2015.

Hall, S. J., Liptzin, D., Buss, H. L., DeAngelis, K. and Silver, W. L.: Drivers and patterns of iron redox
10  cycling from surface to bedrock in a deep tropical forest soil: a new conceptual model, Biogeochemistry, 130(1–2), 177–190, doi:10.1007/s10533-016-0251-3, 2016.

[revised manuscript text omitted]
. ~~Despite this trend, there were overall lower concentrations of reactive metals in the lowland Cg horizon when taking into account both Fe and Al contents (Table 3). E_h values measured in lowland soil during the flooded period are sufficiently low for Fe(III) oxide reduction (Fig. 2d), likely causing the depletion in Fe_o observed here. The relatively low power of Fe_o to predict C concentrations in the lowland positions compared to the upland positions,sreduced and depleted the soils of Fe oxides.~~lowered the capacity for C to accumulate with redox-active Fe oxides.

Even though the lowland soils also had an lower concentrations While the predictive power of- Fe$_o$ diminished across the upland-to-lowland transect, -of Al$_o$ , compared to the upland position became are stronger predictor of C concentrations (FIG/TABLEFig. 4a-b, Table 4). , Consequently, our linear mixed effects model showed that Fe$_o$ served as the strongest measured predictor variable for C in upland soils, yet Fe$_o$ had no predictive power in lowland soils (Fig. 4a). In contrast, our linear mixed effects models show that the strength of the relationship of C with Al$_o$ significantly increaseeds across upland to lowland transitions (Fig. 4a). In contrast to Fe, Al hydroxides are not reducible to a more soluble lower oxidation state. Al hydroxides are thus more likely to accumulate in a dynamic redox environment such as our lowland soils. In fact, we found consistently higher Al$_o$ than Fe$_o$ contents in the lowland soils (Table 3). In similarly dynamic forested floodplain environments, C content was also found to be more strongly correlated with Al$_o$ than Fe$_o$, which was attributed to the formation of stable Al$_3^+$-OM complexes (Darke and Walbridge, 2000). Stronger correlations between Al and C in seasonally flooded mineral soils could be attributed to the overall higher concentrations of Al$_o$ present in the soil, compared to Fe$_o$ (Table 3). Organic matter content in floodplain soils has also been reported to be more strongly correlated with Al$_o$ than Fe$_o$ (Darke and Walbridge, 2000). Stronger correlations between Al and C in seasonally flooded mineral soils could be attributed to the overall higher concentrations of Al$_o$ present in the soil, compared to Fe$_o$ (Table 3). Additionally, high soil OM contents,High OM contents, typically found in wetland soilsas found in our lowland soils, have also been found to stabilize Al$_3^+$-OM complexes by inhibiting crystallization of Al into more crystalline, and less reactive can inhibit Al oxides crystallization if Al$^{3+}$ if incorporated into Al-OM complexes (Darke and Walbridge, 2000; Borggaard et al., 1990). These Al-OM complexes result in more poorly crystalline, highly reactive, Al oxides. Thus, mMineral protection of CC accumulation in our seasonally reduced flooded mineral wetland soils may partly thus depend on non-reducible Al oxidesnon-reducible, poorly crystalline Al$_3^+$-OM complexes.

Together, our these results indicate clearly illustrate the that the relative importance ofr roots, mineral composition and redox conditions on that 
[revised manuscript text omitted]
 | 3.9$^b$ ± 0.1 | 14$^a$ ± 3.8 | 48$^b$ ± 17 | 5.43$^a$ ± 0.2 | 51$^a$ ± 10 | 1.2$^a$ ± 0.3 | 3.3$^a$ ± 0.4 |
| **Lowland** | | | | | | | |
| Cg | 1.9$^b$ ± 0.5 | 13$^a$ ± 3.7 | 2$^b$ ± 0.6 | 5.29$^a$ ± 0.1 | 66$^a$ ± 9 | 1.0$^a$ ± 0.3 | 2.6$^{ab}$ ± 0.5 |
| Cg | 0.36$^a$ ± 0.02 | 7.3$^a$ ± 4.9 | 0.7$^a$ ± 0.5 | 5.37$^a$ ± 0.1 | 70$^a$ ± 9 | 2.9$^a$ ± 0.7 | 1.0$^b$ ± 0.2 |

Letter designations indicate significant differences among similar horizons. Compared within surface (A horizons), intermediate (B horizons), and subsurface (C horizons) horizons were determined by ANOVA followed by Tukey's HSD results based on complimentary horizon basis. Different letter designations indicate a p-value of < 0.05.

Table 4 Fixed effect parameters from the linear mixed models along the upland-to-lowland transects

| Variable | Degrees of freedom | Regression Coefficient ± standard error | F - value | Landscape Prob > F | Horizon Prob >F |
|---|---|---|---|---|---|
| **Upland** | | | | | |
| Root Biomass | 17 | $0.15 \pm 0.09$ | 13.31 | <0.01 | NS |
| $Fe_o$ | 17 | $0.37 \pm 0.15$ | 17.31 | <0.001 | NS |
| $Al_o$ | 17 | $0.31 \pm 0.09$ | 10.76 | <0.01 | <0.05 |
| Clay | 17 | $0.39 \pm 0.23$ | 8.56 | <0.01 | <0.05 |
| $E_h$ | 17 | $0.24 \pm 0.31$ | 2.86 | NS | <0.05 |
| **Transition** | | | | | |
| Root Biomass | 18 | $0.05 \pm 0.07$ | 21.81 | <0.001 | <0.05 |
| $Fe_o$ | 18 | $-0.05 \pm 0.08$ | 2.86 | NS | <0.0001 |
| $Al_o$ | 18 | $0.14 \pm 0.14$ | 15.57 | <0.001 | <0.001 |
| Clay | 18 | $0.24 \pm 0.07$ | 1.68 | NS | <0.0001 |
| $E_h$ | 18 | $-0.08 \pm 0.08$ | 0.32 | NS | <0.0001 |
| **Lowland** | | | | | |
| Root Biomass | 12 | $0.12 \pm 0.38$ | 11.22 | <0.01 | NS |
| $Fe_o$ | 12 | $-0.01 \pm 0.17$ | 0.45 | NS | <0.01 |
| $Al_o$ | 12 | $0.91 \pm 0.15$ | 137.36 | <0.0001 | NS |
| Clay | 12 | $-0.08 \pm 0.21$ | 0.31 | NS | <0.01 |
| $E_h$ | 12 | $-0.47 \pm 0.23$ | 0.77 | NS | <0.01 |

Model parameters with p-values > 0.05 are denoted as not-significant with the letters NS.

[Figure]

fig01

[Figure]

fig02

[Figure]

fig03

[Figure]

fig03

(a)

[Figure]

(b)

[Figure]

fig04

(a)

[Figure]

(b)

[Figure]

fig04

[Figure]

fig05

[Figure]

fig05

[Figure]

fig06

[Figure]

fig06

---

## Referee Report (RR1)

This manuscript investigated the environmental and biogeochemical factors in controlling $CO_2$ efflux and organic matter composition along upland to lowland transition in seasonally flooded mineral soils. The experiment in this study is well designed, linked soil $CO_2$ emissions and related soil and plant properties in field condition. Some of results produced from this study are valuable, to certain extent, and provide additional data to fill the gap if considering the effects of seasonal flooding and mineral composition on C dynamics were rarely investigated in the field. I think the paper is publishable, but it requires minor revisions. Below, please find a list of comments that would be helpful to consider for revision of this paper.

**General comments:**

1) More recent literatures are needed in the discussion, to make clear what have been done. For example, Page 16 (Lines 15-20), Page 17 (Lines 5-10) the authors cited literatures about contribution of reactive Fe and Al on C accumulation.

2) I would like to see in depth discussion on the physiochemical mechanisms involved in the process of C dynamics with respected to Fe and Al phases in both upland and lowland filed should also be summarized here based on the published literatures. For example, page 16 (line 20-25) presented only reductive dissolution of Fe and Al phases.

3) Further experiment needed on abiotic component (MBC or community structure) and will add values on the obtained data and comprehensive understanding on processes. This would certainly help us better understand the role of wetting and drying cycles on GHG ($CO_2$, in this work).

---

## Author Response (AR2)

**COMMENTS BY THE ASSOCIATE EDITOR (AE):**

Dear Dr Keiluweit,

One of the previous reviewer has agreed and satisfied with all the changes, but another reviewer not accepted the revised version to have further comments. Thus I nominate another two and one is relatively positive while the other is critical. Please find the comments and make a reasonable changes and response. My personal concern is to add more details about FT ICR-MS data and correlate with Fe/Al and other varialbles.

Kind regards
Yu

**Authors' response to AE comments:**

Dear Dr. Luo,

Thank you for your willingness to consider a revised version of our manuscript entitled *"Shifting mineral and redox controls on carbon cycling in seasonally flooded mineral soils"* for publication in *Biogeosciences*. We addressed the issues and concerns presented in the two additional reviews, as detailed in our point-by-point response below.

Briefly, as requested by reviewer #4, we added more detailed information on the methods used for FTICRMS analysis and revised the results and discussion sections. We also expanded our discussion of the biogeochemical mechanisms involved as requested by reviewer #3. While we had conducted multiple linear regressions between FTICRMS results and Fe/Al variables as part of our initial data analysis, as suggested in your comment above, we found no significant differences and no discernable trends. We therefore did not report this information.

We did not provide a second response to reviewer #2 (Dr. Aaron Thompson), who indicated that we had successfully addressed all of the comments in his initial review of the manuscript. He rated the scientific significance and presentation quality of the manuscript as "excellent" and recommended "publication as is".

We hope that these changes adequately address your concerns. Thank you for considering our revised manuscript for publication in *Biogeosciences*.

Sincerely,

Marco Keiluweit, on behalf of the co-authors

**RESPONSE TO REVIEWERS' COMMENTS**

Regular font:   original comment by the reviewer
*Italized text:*           response by the authors
*"Italicized quotes":*   revised text segments

**Response to Comments by Referee #3**

Reviewer's Comment: I have revised manuscript 'Shifting Mineral and Redox Controls on Carbon Cycling in Seasonally Flooded Mineral Soils'. This work focuses on an estimation of various factors (Eh, extractable Fe and Al, and root biomass) on the CO2 emission from seasonally flooded forest soils in comparison to never flooded, and was done for the small scale transects. Moreover, the quality composition of SOM was determined in three soil horizons for these transect soils by the spectroscopic method (NEXAFS) and composition of DOC by the Fourier-transform ion cyclotron resonance mass spectrometry. During my revision, I found out that the manuscript has many drawbacks, especially in the methodological part and following results interpretation.

My main concerns are: i) statement that upland soils are not affected by water fluctuations (actually, they are affected, which I can see well from the Eh values), ii) interpretation of the composition of SOM by the methods, which can only provide qualitative estimation, but not quantitative; iii) discussion of the data which are not statistically significant, iv) absence of any soil microbiological analyses, which can explain the differences for CO2 emission in various seasons.

*Authors' response:*
*We thank the reviewer for these constructive comments and have taken the following steps to address them.*

*i) We agree with the reviewer that the upland soil undergoes seasonal changes in moisture that are reflected in the redox data. We have revised the description of the water table dynamics at the three landscape positions accordingly.*

*Author's changes: (Page 7, Ln 5-10):* "The upland position does not experience flooding in a typical rainfall year, i.e., the water table does not rise above the soil surface. The transition position is located on the edge of the wetland, which typically does not get flooded in an average rainfall year but is under the influence of significant water table rise. The lowland position is in the lowest point of the transect and is flooded for several months throughout the year."

*ii) We revised the description of the results to reflect the semi-quantitative nature of the mass spectrometry and spectroscopy results. We would further like to note that the methods used here are semi-quantitative. They allow to assess shifts in the relative abundance of different functional groups (C NEXAFS) or compound classes (FTICRMS) as is done in all other studies relying on these techniques. This is also the case for almost all other chemical methods for the*

*characterization of SOM (13C NMR, py-GC/MS, FTIR, XPS, etc), where natural abundances are commonly reported.*

*Authors' changes: (Page 16, Ln 12-16): "The composition of water extractable OM was similar across the transect (p-value > 0.05), but some general trends were noticeable. Both the modified aromaticity index ($AI_{mod}$) and the average molecular weight of the detected compounds showed gradual increases across the upland-to-lowland transitions in the surface horizons (Fig. 6a, Table S6). Paralleling that change, the relative contributions of lignin increased (+7%) and that of lipids decreased (-11%) moving from the upland to the lowland position (Fig. 6b, Table S5). In the subsurface horizons, however, both $AI_{mod}$ and average molecular weight showed little changes (Fig. 6a, Table S6), while the relative abundance of lignin increased (+9%) and that of lipids decreased (-11%)."*

*(iii) We appreciate the reviewer's concern. Throughout the manuscript, we took greatest care to only report and discuss significant differences. The only data where we made an exception is the FTICRMS results. We decided to discuss the FTICRMS results because there are clear trends across the transect that are consistent with other observations (e.g., C concentration and functional groups chemistry) and was thus deemed of greater interest to the general public. To address the reviewer's comment, we revised the results section to clearly communicate the limitations in the data in the most transparent fashion possible.*

*Authors' changes: (Page 16, Ln 16-24): "To assess changes in oxidation state and molecular weight of compounds more readily available for microbial respiration, water extracts of all samples were analyzed by FT-ICR-MS (Fig. 6a-b, Fig. S1-3, Table S5, Table S6). The composition of water extractable OM was statistically indistinguishable across the transect (p-value > 0.05), but some general trends were noticeable. Both the modified aromaticity index ($AI_{mod}$) and the average molecular weight of the detected compounds showed gradual increases across the upland-to-lowland transitions in the surface horizons (Fig. 6a, Table S6). Paralleling that change, the relative contributions of lignin increased (+7%) and that of lipids decreased (-11%) moving from the upland to the lowland position (Fig. 6b, Table S5). In the subsurface horizons, however, both $AI_{mod}$ and average molecular weight showed little changes (Fig. 6a, Table S6), while the relative abundance of lignin increased (+9%) and that of lipids decreased (-11%)."*

*(iv) We agree with the reviewer that microbial processes are at the center of soil carbon cycling, and that differences in microbial mineralization rates are the cause for the differences in CO2 emissions across the transect. This study aimed to disentangle the biogeochemical processes that cause C accumulation and loss (through microbial mineralization) in seasonally flooded soils, and thus we focused on the coupled biogeochemical measurements rather than detailed microbial biology. While we did not include direct measures of microbial biomass, community composition or function, we did measure microbial activity ($CO_2$ production) and metabolic potential (redox potential). We fully agree that more detailed analyses of microbial community composition and function would be an interesting endeavor for the future.*

General comments:

Reviewer's Comment: The article focuses on the C cycling in seasonally flooded mineral soils, however, had looked only on the CO2 emission, what about the methane? How strong can be the contribution of this gas to C flux out from the soil? What is known from the literature?

*Authors' response: We addressed this concern briefly in the methods section (previously: Page 7, Ln 11-19). Briefly, $CO_2$ emissions are found to be orders of magnitude greater than CH4 emissions in comparable wetlands. We expanded this section as follows and moved it to the introduction section.*

*Authors' changes: (Page 6, Ln 9-14): "Based on other reports for comparable sites (Holgerson, 2015; Kifner et al. 2018), we expected methane production within these seasonal wetlands. However, $CO_2$ emissions were at least 15-times greater than methane production in those sites. While we acknowledge the disproportionate potency of methane as a climate-active greenhouse gas, this study aimed to determine the environmental and biogeochemical factors influencing C accrual or depletion in across the upland-lowland transition. We thus focused our monitoring efforts on quantitatively more important $CO_2$ emissions as the predominant C loss pathway."*

Reviewer's Comment: Introduction in general: you have provided some information about the knowledge gaps, but did not explain any mechanisms for the processes going in the periodically flooded soils, and what is unclear. Please, provide more details.

*Authors' response: We revised the following paragraph in the introduction to crystallize the knowledge gap. We further provided more detail on specific mechanisms as detailed in the specific comments below.*

*Authors' changes: (Page 5, Ln 15-21): "Water saturation thus likely governs C cycling in seasonally flooded soils through its combined impact on oxygen availability, root dynamics and mineral composition; but how the relative contribution of these biogeochemical controls vary across spatial and temporal gradients is still unknown. A recent study along hillslope transects in tropical forest soils representing an oxygen gradient (Hall and Silver, 2015), for example, found that a combination of Fe (II) (a proxy for reducing conditions), fine root biomass, and total Fe and Al concentrations explained the most variation of surface soil C content. How the relationships between C content and important biogeochemical controls differ in systems that are subject to seasonal flooding is still in question, especially with depth (Barcellos et al., 2018)."*

Reviewer's Comment: Besides, you have a lack of information about microbial driven processes. From the results, I can see that you did not touch this topic, but actually, C fate in the soil is driven by microorganisms.

*Authors' response:*
*(see IV above): As mentioned above, this study focuses on the environmental and biogeochemical controls on C in seasonal wetlands. Specifically, we examine how environmental and biogeochemical controls influence microbial OM decomposition and CO2 production. While we do not include measures of microbial biomass, diversity or function, we do assess microbial activity (as evidenced by CO2 efflux) and metabolic potential (as evidenced by redox potential,*

*and OM transformation (as evidenced by FT ICR MS and NEXAFS analyses). We do agree that more detailed microbial analyses of the linkage between seasonal flooding and microbial activity and function would be an interesting endeavor for the future.* [TM-(1]

Reviewer's Comment: Moreover, you should point I the Introduction the reason why you focus only on the CO2, and not on methane estimations.

*Authors' response: Please see our previous response and changes above.*

Reviewer's Comment: No information is presented about the effect of temperature on the processes you are going to study.

*Authors' response: We revised the introduction to add more information regarding the effects of soil temperature on carbon cycling in soil.*

*Authors' changes: (Page 4, Ln 1-4): " Temperature and soil moisture are principle controls on C cycling in soils (Lloyd and Taylor, 1994; Wang et al., 2014). Temperature regulates biological and chemical reaction rates and thus regulates the rate at which decomposition of soil organic matter can occur (Davidson and Janssens, 2006). However, water saturation is a critical driver of organic matter (OM) decomposition processes in seasonally flooded systems (Neckles and Niell, 1994)."*

Reviewer's Comment: Hypothesis1 – it is not clear: 1) capacity of which minerals do you mean? 2) greater C accumulation for which period? 3) what does macromolecular and chemically-reduced OM mean?

*Authors' response:*
*1) We specified the reactive minerals under consideration.*

*Authors' changes: (Page 6, Ln 7): "… lower capacity of Fe/Al (hydr)oxides to protect OM,…".*

*2) C accumulation here refers to the amount of C accumulated during the formation of lowland versus upland soils in this position.*

*Authors' changes: (Page 6, Ln 6-8): "We hypothesized that seasonally reduced conditions upon flooding in lowland positions will result in lower $CO_2$ efflux, greater C accumulation, lower capacity of Fe/Al (hydr)oxides to protect OM, and a selective preservation of macromolecular or chemically-reduced OM compared to the upland position."*

*3) We defined both terms in the introduction (Page 4, Ln 8). A macromolecule is "a molecule of high relative molecular mass, the structure of which essentially comprises the multiple repetition of units derived, actually or conceptually, from molecules of low relative molecular mass." (IUPAC Gold Book definition). Chemically-reduced compounds are in reference to organic carbon compounds with lower C oxidation states (Page 4, Ln 11-13).*

Reviewer's Comment: 4) what are specific sites? Forest as well as grassland soils can be under periodic flood conditions in various climates and can be developed on different parent materials.

Authors' response: *The specific sites are described in a detailed site description as part of the methods section (Page 6, Ln 17 – Page 7, Ln 8).*

Reviewer's Comment: Moreover, you did not provide any information on how the quality of accumulated organic compounds in flooded condition is different from upland.

Authors' response: *We are slightly confused by this comment. We present an in-depth molecular characterization of the organic matter that has accumulated in lowland and upland positions using two advanced analytical techniques, NEXAFS and FTICRMS.*

Reviewer's Comment: There are many works done for the rice soils, for example, which are also under the pulse soil moisture conditions.

Authors' response: *We agree that there is a growing body of literature on C cycling in rice paddies. While we discuss and cite relevant publications (see excerpt below), the specific objective of this study was to investigate the seasonal dynamics in natural, temperate, seasonally flooded wetlands of the northeastern US. They are distinctly different from rice paddies with respect to seasonal temperature, precipitation, climate and soil type.*

*Favre, F., Tessier, D., Abdelmoula, M., Génin, J. M., Gates, W. P. and Boivin, P.: Iron Reduction and Changes in Cation Exchange Capacity in Intermittently Waterlogged Soil, Eur. J. Soil Sci., 53(2), 175–183, doi:10.1046/j.1365-2389.2002.00423.x, 2002.*

*Hanke, A., Sauerwein, M., Kaiser, K. and Kalbitz, K.: Does Anoxic Processing of Dissolved Organic Matter Affect Organic–Mineral Interactions in Paddy Soils?, Geoderma, 228–229, 62–66, doi:[10.1016/j.geoderma.2013.12.006](10.1016/j.geoderma.2013.12.006), 2014.*

*Kögel-Knabner, I., Amelung, W., Cao, Z., Fiedler, S., Frenzel, P., Jahn, R., Kalbitz, K., Kölbl, A. and Schloter, M.: Biogeochemistry of Paddy Soils, Geoderma, 157(1–2), 1–14, doi:10.1016/j.geoderma.2010.03.009, 2010.*

Authors' changes: *(Page 18, Ln 10-17): "The diminished importance of $Fe_o$ in C accumulation in our seasonally flooded lowland soils is consistent with the loss of reactive Fe phases observed in forest soils (Fiedler and Kalbitz, 2003; Zhao et al., 2017) and rice paddy soils (Favre et al., 2002; Kögel-Knabner et al., 2010; Hanke et al., 2014). High concentrations of organic C in rice paddy surface soils drives the reduction and dissolution of redox-active minerals, such as Fe(III) oxides, which is subsequently translocated vertically down the soil profile (Kögel-Knabner et al., 2010, Chen et al., 2017). In our sites, we found a noticeable, yet insignificant, increase in $Fe_o$ contents in the lowland $C_g$-horizons (Table 3), which is likely a reflection of these vertical transport processes of soluble or colloidal Fe phases into the subsurface horizon, where they may reprecipitate during drained periods (Kögel-Knabner et al., 2010; Hanke et al., 2014)."*

Reviewer's comment: Materials and method: not clear how far where the plots from each other; is it possible to know the approximate age of these relief form (I mean they were permanently here or formed only when the experimental station was established?); how many soil cores were collected and how from each horizon and from each site? How many samples were collected for root analysis from each site and each landscape position?

*Authors' response: Fig. 1 depicts the distance between the individual positions, which ranged from 3 to 5 meters. The replicate transects are within approximately 100 meters of each other. The two sites (South Deerfield and Amherst) are approx. 9 mi apart. They are indistinguishable in terms of vegetation, climate and soil types. In fact, our analyses found no statistical differences between the two sites. We updated the site description to include this information.*

*Authors' changes: (Page 7, Ln 4-5): "Fig. 1 depicts the distance between the individual positions, which ranged from 3 to 5 meters. The replicate transects are within approximately 100 meters of each other."*

*We don't know the exact age of the relief form, but the wetlands are in natural depressions in areas unaffected by agricultural use. They can thus be considered historic seasonal wetlands, as evidenced in the development of horizons characteristic of seasonal wetlands. This pedogenic process will take 100s of years, reflecting the general age of the wetlands.*

*Authors' changes: (Page 8, Ln 1-7): "Soil samples were collected using hand augers in each of the six replicate wetlands along the defined transects that included the three landscape positions (upland, transition, and lowland). In each landscape position, we collected soil cores from designated horizons, resulting in six replicate cores for each horizon and position. Horizons in the upland position were classified as A (0-25 cm), B (25-55 cm), and C (55-84+ cm) horizons; in the transition position as A (0-28 cm), C (28-48 cm), and $C_g$ (48-69+); and in the lowland position as A (0-25 cm), C (25-35 cm), and $C_g$ (35-68+ cm) (Soil Survey Staff, 1999) (Fig. 1a)."*

*Root biomass samples were collected in parallel with the selected representative transects as soil samples. We describe the method of collection in the revised manuscript (Page 8, Ln 14-18).*

*Authors' changes: (Page 8, Ln 15-16) "The cores were taken at 0-20, 20-40, and >40 cm and resulted in six cores per landscape position horizon (e.g., upland A-horizons n = 6)."*

Reviewer's Comment: Fourier-transform ion cyclotron resonance mass spectrometry – method is not so used in soil science and can confuse readers, please provide additional information whether a liquid sample is introduced as a liquid for the measurement, or prepared additionally (dried or by any other way).

*Authors' response: We have revised the section to include how the samples were prepared and introduced to the instrument. Briefly, soil samples were first extracted in ultrapure DI-H2O using one gram of soil and 10 mL of DI-H2O (1:10). After extraction, samples were filtered through 0.2-μm syringe filters. 100μl of the filtrate was then added to 200 μl of methanol (1:2) and injected directly (in liquid phase) onto the instrument.*

*Authors' changes: (Page 9, Ln 10-19) "To determine the composition of bioavailable compounds that can potentially be used in microbial respiration (<600Da, Logue et al., 2016), water extracts of soil samples were collected on a 12 Tesla Bruker SolariX Fourier-transform ion cyclotron resonance mass spectrometer located at Environmental Molecular Sciences Laboratory (EMSL), a Department of Energy Biological and Environmental Research (DOE-BER) national user facility located in Richland, WA. Soil samples were extracted with ultrapure DI-$H_2O$ using one gram of soil and 10 mL of DI-$H_2O$ (1:10). The samples were sealed in 15 mL conical tip tubes and shaken for one hour. Samples were then centrifuged and filtered using syringe-filters. 100 µL of the filtrate was then added to 200 µL of methanol (1:2) and injected directly (in liquid phase) onto the instrument. A standard Bruker electrospray ionization (ESI) source was used to generate negatively charged molecular ions; samples were then introduced directly to the ESI source. The instrument was externally calibrated to a mass accuracy of <0.1 ppm weekly using a tuning solution from Agilent".*

Reviewer's Comment: For the results: there are many uncertainties in the method which you used to estimate the portion of various substances classes based on the FT-ICR-MS analysis, including ionization efficiencies, mobile phase composition, data acquisition parameters. Thus, I think it is better if you will re-do the figure 6, like it is originally advised in the article by Kim et al., 2003.

*Authors' response: In response to the reviewer's comment, we added Van Krevelen Diagrams (VKD) for each sample in the SI (Fig. S1-3) and kept Figure 6 as is in the main manuscript. While FTICR MS is not a quantitative approach due to ionization competition during ESI process, all samples were run under the same exact conditions. As such comparison between samples is valid. One of the most common ways of visualizing FTICR MS data is through VKD (as suggested by the reviewer). This is a suitable approach when dealing with a handful of samples, but in this case we had roughly 70 samples, making it hard to visualize each sample through a VKD. Another way to use FTICR MS data is to group the peaks on the VKD into different compound classes based on their location on the VKD, for example lipid –like compounds are known to have low O/C and high H/C. After grouping all peaks based on their compound class, you can calculate relative abundance of each compound class. Since all samples were treated exactly the same, then this data can be used in a semi quantitative way to compare between different samples. This approach is highly used when dealing with a large number of samples. We followed the same approach in Figure 6 where soil samples were divided into three groups: upland, transition and lowland. Then each group was separated into three horizons. Only compounds/peaks that were unique to each horizon were then included in the calculation. We believe this approach works great for this data type especially to be able to see difference between samples, especially when it is hard to visualize 60 VKDs at the same time.*

Reviewer's Comment: Another suggestion can be to determine at least some components by the true analytical methods in the bulk soil samples and not in the DOC (for example lignin and carbohydrates contents).

*Authors' response: The authors agree that analysis of bulk SOM composition is pertinent to this study, which is why we have conducted C (1s) NEXAFS analyses (Page 8, Ln 20-24). This type of analyses is optimal for measuring C functional group characteristics as it is performed on bulk*

*soil samples. Results of these analyses are reported in section 3.5 and discussed throughout the discussion section. Additionally, Fig. 5 shows the individual spectra and average NEXAFS spectra for the landscape positions and horizons. Peak deconvolution is then used to determine relative abundances of C functional groups within each bulk soil sample. Full relative abundance of these functional groups are reported in Table S4.*

Reviewer's Comment: Try to put conclusions at the end of each resulting part, summarizing the findings.

*Authors' response: We have added short conclusions to sections 3.1 and 3.3 in the results section. However, we feel that adding further conclusion remarks to sections 3.2, 3.4 and 3.5 would be redundant.*

*Authors' changes:*
*(Section 3.1, Page 13, Ln 13-15): "In sum, the rise in seasonal water table in lowland positions during the growing season was accompanied by decreased $E_h$ and $CO_2$ fluxes compared to the upland positions. As the water table dropped during the growing season $E_h$ and $CO_2$ fluxes increased markedly."*
*(Section 3.3, Page 14, Ln 3-5) "In sum, $CO_2$ emissions in the upland position were most strongly correlated to soil temperature, while water table depth and VMC correlated more strongly with $CO_2$ fluxes in the lowland position during the growing season."*

Reviewer's Comment: Everywhere – when you speak about C in the soil, please use 'content' (g kg-1 soil) and 'concentration' when you speak about DOC.

*Authors' response: We have updated the terminology in the manuscript to reflect this suggestion.*

Reviewer's Comment: Table 1 – Actually shows no differences between CO2 effluxes, is it right?

*Authors' response: We appreciate the reviewer pointing out this discrepancy. We clarified that $CO_2$ effluxes were significantly different among the three landscape positions only when normalized to C content (revised Table 1).*

Reviewer's Comment: Fig. 1 and Fig. 2 – from the Fig 1 I can see, that actually all horizons C for all soils are actually all the time underwater. I have than a question: how did you measure the Eh? And how did you collect soils samples?

*Authors' response:*
*We apologize for the confusion. The upland position, including the C horizon, are entirely drained during the dry season. We revised Fig. 1a to more accurately depict the water table dynamics in the upland landscape position. Moreover, we corrected Fig. 2b. In this figure, we had plotted the water table to be at 50 cm depth during the dry period, even though the water table had dropped below the measuring gauge. The measuring gauges were installed at 50 cm depth, thus when the water table was at or below 50 cm it was recorded at 50 cm and that was shown in Fig. 2b (October through December). This gave the false impression that the C-*

*horizons in the upland position are saturated during these months. To address this mistake, we removed the data points from Fig. 2b and noted that the water table dropped below this mark (Page 7, Ln 18-19).*

*Authors' changes* *(Page 7, Ln 18-19): "Water table fluctuations were monitored using slotted PVC pipes installed to depths of 50 cm, therefore water table depths below 50 cm were undetectable and not reported (Fig.2b)."*

*(Page 31, Ln 12-14): "Missing data points for the upland position in panel (b) are due to the water table dropping below the measuring gauge installed at a depth of 50 cm and is denoted with asterisks (*)."*

*As described in the methods section (Page 7, Ln 19-25), we measured Eh, we installed redox probes at each location to depths of 15, 30, and 45cm in triplicate in each landscape position of the six sites (Page 10, Ln 4-8). The redox probes were left in place for the entire year-long study and measured in parallel with $CO_2$, VMC and water table fluxes.*

Reviewer's Comment: Moreover, Upland soil is also under the effect of water table rise (I can see it very well from the Eh values).

*Authors' response:* *We agree that the upland position does see a rise in water table depth during the fall recharge period, and that the subsurface horizons are affected by the fluctuations in moisture content. However, we do not make this claim in the manuscript, rather we are comparing flooded soils in the lowland positions which are submerged under water to their upland counterparts which are not submerged at any point throughout the year.*

*(Page 7, Ln 5-8): "The upland position is in a forested landscape, approximately five meters away from the edge of the wetland, which does not undergo any flooding. The transition position is located on the edge of the wetland, which typically does not get flooded in an average rainfall year but is under the influence of water table rise. The lowland position is in the lowest point of the wetland and is flooded for several months throughout the year."*

Reviewer's Comment: Fig 3 is not discussed in the result part.

*Authors' response:* *We greatly appreciate the reviewer for noticing the missing section from our manuscript. This results section had been inadvertently deleted from the manuscript. This section has been added back into the manuscript.*

*Authors' changes:* *(Page 13, Ln 16 – Page 14, Ln 16):*
*"3.2 Control on $CO_2$ fluxes*
*To determine which of the above environmental parameters best predict soil respiration across the hydrological gradient, we conducted a series of regression analyses (Fig. 3a-d). Regression analyses were carried out for subsets of the data representing the (i) full year, (ii) growing season or (iii) non-growing season (Table 2).*
*        Soil temperature. The strength of the relationship between $CO_2$ flux and soil temperature, as expressed by how well the data can be described using the Arrhenius equation (Sierra, 2012),*

*decreased along the upland-to-lowland transect (Fig. 3a). Soil temperature explained the most variance of $CO_2$ fluxes in the upland positions throughout the full year (r = 0.72, p < 0.001) and the growing season (r = 0.62, p < 0.001) (Table 2). Comparing the three landscape positions, soil temperature explained the least variation of $CO_2$ fluxes in all cases in the lowland position, especially in the growing season (r = 0.45, p < 0.001).*

*Soil moisture. As the relationship between $CO_2$ flux and soil temperature became weaker, that between $CO_2$ flux and water table depth gradually became stronger along upland-to-lowland transitions. $CO_2$ flux and water table depth (Fig. 2b) were significantly negatively correlated in the lowland positions in the full, growing season and non-growing season time periods (Table 2). The strongest correlation between water table depth and $CO_2$ flux occurred in the lowland position during the growing season (r = -0.55, p < 0.001), where it had a stronger relationship with $CO_2$ flux than soil temperature. Similarly, VMC and $CO_2$ fluxes were negatively correlated in the lowland position (r = -0.51, p < 0.001), with VMC showing a stronger relationship with $CO_2$ flux than soil temperature during the growing season (Fig. 2c) Soil redox potential, In keeping with a strong relationship between moisture and respiration at the transition and lowland positions, $E_h$ was also most significantly correlated with $CO_2$ at the transition and lowland positions (Fig. 2d). $E_h$ was a comparable predictor for $CO_2$ flux in both the lowland (r = 0.40, p-value <0.001) and transition (r = 0.41, p-value <0.001) positions during the growing season, but had no correlation with $CO_2$ flux in the upland position (Table 2). The strong correlations between $E_h$ and $CO_2$ emissions were primarily limited to the lowland position.*

*In sum, $CO_2$ emissions in the upland position were most strongly correlated to soil temperature, while water table and VMC correlated more strongly with $CO_2$ fluxes in the lowland position during the growing season."*

Reviewer's Comment: Discussion part (4.3) Discussion about fig. 5 does not support by statistical analyses (near all differences are not statistically significant). I suggest that you try to test differences between the horizons. Besides, data on figure 6 are qualitative and not quantitative (see comments above).

*Authors' response: The statistical analyses regarding the data in Fig. 5 are reported in Table S4. As the reviewer suggested, we did test differences based on a horizon basis.*

Reviewer's Comment: Part 4.4. This is basically a repetition of the discussion above, and suggest to include these references into the discussion part to make it deeper in sense.

*Authors' response: Section 4.4 explores how any moisture and temperature changes due to climate change may affect the dynamics within these vulnerable ecosystems. We feel these important implications are the core message of the paper and should be included.*

Specific comments

Reviewer's Comment: P3L11 '' seasonally flooded soils are metabolically more active'' – please be more precise here, what do you mean?

*Authors' response: We revised this sentence to address the reviewer's concern.*

*Authors' changes: (Page 3, Ln 10-12): "This is surprising given that seasonally flooded soils are characterized by greater microbial activity than permanently flooded wetlands, resulting in significantly greater greenhouse gas emissions (Kifner et al. 2018)."*

Reviewer's Comment: P4L5 Forgot oxidation state - SO4 2-

*Authors' response: We have revised this sentence accordingly.*

*Authors' changes: (Page 4, Ln 21): "Further, once oxygen is depleted, microbes rely on alternative terminal electron acceptors ($NO_3^-$, $Mn^{4+}$, $Fe^{3+}$, $SO_4^{2-}$)…"*

Reviewer's Comment: P3L25 – P4L12 Please be more precise here: the activity of which enzymes are inhibited;

*Authors' response: We have revised the sentence for clarity of which enzymes we are referring to.*

*Authors' changes: (Page 4, Ln 5-8): "The resulting oxygen limitations inhibit the activity of oxidative enzymes, such as phenol oxidase or peroxidase (Freeman et al., 2001; Keiluweit et al., 2016), which catalyze the depolymerization of higher-molecular weight OM compounds (i.e., macromolecules) into smaller, assimilable compounds (Megonigal et al. 2003)."*

Reviewer's Comment: which exactly levels of Eh are usually observed in the seasonally flooded soils? This is important to write because the various electron-donor system is active for various Eh values; how does soil C content effect on the Eh conditions?

*Authors' response: Seasonal variations in Eh are given in Fig. 2d and Table S2. Upland Eh ranges fall between approximately 390 and 500 mV during the growing season, and between 595 and 618 mV during the non-growing season. Transition Eh ranges fall between approximately 190 and 240 mV during the growing season, and between 480 and 580 mV during the non-growing season. Lowland Eh ranges fall between approximately 70 and 90 mV during the growing season, and between 415 and 450 mV during the non-growing season. There were no significant relationships found between C content and Eh in our systems.*

Reviewer's Comment: P8L2 What does spline function mean?

*Authors' response: An equal-area quadratic spline function is a mathematical function to estimate properties in soil, which fits curves piecewise throughout the entire soil profile based on given data points. The curves ensure there is equal areas to the left and right of the spline curve. This method enables us to use non-horizon based data points and estimate their values for given horizons. This method has been extensively used in soil science since first applied by Bishop et al. (1999), and later refined by Malone et al. (2009). The below references further explain how equal area spline functions can be used in soil science. We have revised this sentence with references.*

*Odgers, N. P., Libohova, Z. and Thompson, J. A.: Equal-area spline functions applied to a legacy soil database to create weighted-means maps of soil organic carbon at a continental scale, Geoderma, 189–190, 153–163, doi:10.1016/j.geoderma.2012.05.026, 2012.*

*Malone, B.P., McBratney, A.B., Minasny, B., and Laslett, G.M.: Mapping continuous depth functions of soil carbon storage and available water capacity, Geoderma, 154(1), 138-152, doi.org/10.1016/j.geoderma.2009.10.007, 2009.*

*Bishop, T.F.A., McBratney, A.B., and Laslett, G.M.: Modelling soil attribute depth functions with equal-area quadratic smoothing splines, Geoderma, 91(1), 27-45, doi.org/10.1016/S0016-7061(99)00003-8, 1999.*

*Authors' changes: (Page 8, Ln 17-19): "The initial values of root biomass were used to estimate biomass values for each soil horizon using an equal-area quadratic spline equation (Malone et al., 2009; Spline Tool v2.0, ASRIS). Mean $E_h$ values for each soil horizon were also estimated using the equal-area quadratic spline equation (Malone et al., 2009)."*

Reviewer's Comment: P8L23 Please, provide filter size for the DOC filtration.

*Authors' response: Water extractable OM samples were filtered using 0.2-$\mu$m syringe filters.*

*Authors' changes: (Page 9, Ln 15): "Samples were then centrifuged and the supernatant filtered using 0.2-$\mu$m syringe filters."*

Reviewer's Comment: P11L17-19 You can delete this sentence and start directly with results.

*Authors' response: We have removed this sentence.*

Reviewer's Comment: P11L20-P12L4 Please do not repeat data presented in the table, as well as the name of the stat. test you can remove, it is given in the table captions and in the mm section. Please, correct everywhere in the result section. Moreover, write only about significant results in the entire article, unless the goal of the paper is to show the absence of significance. Please, correct everywhere in the result section.

*Authors' response: We thank to reviewer for these suggestions and have removed the names of the stats tests from the results section text. In regards to the discussion of non-significant results in the article, the only data we refer to that were not significant was the FT-ICR-MS data. We have previously explained the limitations of the statistical analyses (point iii in initial general comments) and have attempted to present these limitations in a transparent fashion to the reader: Page 16, Ln 11-13: "The composition of water extractable OM similar across the transect (p-value > 0.05), but some general trends were noticeable.".*

Reviewer's Comment: P14L1-P14L16 from Table S4, I can see that only the second horizon was different from for the landscape positions, thus, please, delete description about insignificant

results. It is enough to write that no significant differences were found for other horizons or functional groups.

*Authors' response: Table S4 shows significant differences for relative abundances of aromatic, aliphatic, carboxylic C functional groups between the three landscape positions on a horizon-basis in the surface horizons and second horizon groups. When insignificant trends were mentioned we addressed them as such.*

Reviewer's Comment: Moreover, I do not see the reasons to present Fig 5 in the main result part, because it does not show any significant results. Please transfer to the supplementary materials.

*Authors' response: As noted above, there are significant differences among the relative abundances of C functional groups shown in the NEXAFS spectra presented in Fig. 5. Therefore, we would respectfully keep this figure in the main part of the manuscript.*

Reviewer's Comment: P15L1-2 Please, delete this sentence. Start with the discussion directly.

*Authors' response: We have removed the suggested sentence.*

Comment: This manuscript investigated the environmental and biogeochemical factors in controlling $CO_2$ efflux and organic matter composition along upland to lowland transition in seasonally flooded mineral soils. The experiment in this study is well designed, linked soil $CO_2$ emissions and related soil and plant properties in field condition. Some of results produced from this study are valuable, to certain extent, and provide additional data to fill the gap if considering the effects of seasonal flooding and mineral composition on C dynamics were rarely investigated in the field. I think the paper is publishable, but it requires minor revisions. Below, please find a list of comments that would be helpful to consider for revision of this paper.

*Authors' response: We thank the reviewer for the constructive comments have taken the suggested steps to improve the discussion section of our manuscript.*

**General comments:**

1) More recent literatures are needed in the discussion, to make clear what have been done. For example, Page 16 (Lines 15-20), Page 17 (Lines 5-10) the authors cited literatures about contribution of reactive Fe and Al on C accumulation.

*Authors' response: We thank the reviewer for suggesting more discussion around recent literature to ensure we are clearly identifying what is new and exciting about our study. We have included more discussion in sections 4.1, 4.2 and 4.3 in connection to prior literature in similar systems. However, there are limitations to comparing other study sites, such as wetlands in the Southeastern US and rice paddies, as they differ in climate, vegetation and parent material.*

*Authors' changes:*
*(Section 4.1, Page 17, Ln15-17): "In contrast, $CO_2$ fluxes along a forested wetland gradient in the Southeastern US showed a lesser seasonal convergence (Krauss and Whitbeck, 2012), as the average annual air temperature of this study site is 7 degrees Celcius above the annual average in the Northeastern US."*

*(Section 4.2, Page 18, Ln 17-25): "The diminished importance of $Fe_o$ in C accumulation in our seasonally flooded lowland soils is consistent with the loss of reactive Fe phases observed in forest soils (Fiedler and Kalbitz, 2003; Zhao et al., 2017) and rice paddy soils (Favre et al., 2002; Kögel-Knabner et al., 2010; Hanke et al., 2014). High concentrations of organic C in rice paddy surface soils drives the reduction and dissolution of redox-active minerals, such as Fe(III) oxides, which is subsequently translocated vertically down the soil profile (Kögel-Knabner et al., 2010, Chen et al., 2017). In our sites, we found a noticeable, yet insignificant, increase in $Fe_o$ contents in the lowland $C_g$-horizons (Table 3), which is likely a reflection of these vertical transport processes of soluble or colloidal Fe phases into the subsurface horizon, where they may reprecipitate during drained periods (Kögel-Knabner et al., 2010; Hanke et al., 2014)."*

*(Section 4.3, Page 20, Ln 17-21): "A study of rice paddy soils showed that reductively dissolved Fe (hydr)oxide coatings on vermiculite clay surfaces were re-precipitated on mineral surfaces*

*upon re-oxygenation during water table drawdown (Favre et al. 2002). Therefore, it is likely that seasonally reduced Fe (hydr)oxides are transported down the soil profile and then re-precipitated in the subsurface soil horizon during water table drawdown (Kögel-Knabner et al., 2010)—trapping dissolved, partially-oxidized, lignin-derived OM also leaching down the profile and so resulted in the accumulation of relatively-oxidized OM."*

2) I would like to see in depth discussion on the physiochemical mechanisms involved in the process of C dynamics with respected to Fe and Al phases in both upland and lowland filed should also be summarized here based on the published literatures. For example, page 16 (line 20-25) presented only reductive dissolution of Fe and Al phases.

*Authors' response: We thank the reviewer for this suggestion and refer to our improved discussion of the role of Fe and Al phases in C retention in both reducing and oxidizing soils in the introduction (Page 5, Ln3-14) and discussion sections 4.2 (Page 18, Ln 17 – Page 19, Ln 16) and 4.3 (Page 20, Ln 16-21). Specifically, we point out how reducing conditions limit retention of C by Fe in the lowland sites compared to upland soils, due to reductive dissolution of Fe(III) oxides during anaerobic periods (Page 5, Ln 6-11 and Page 18, Ln 19-21). Conversely, we discuss how Al hydroxides are non-reducible and thus act as the primary sorbent for OM in flooded soils (Page 5, Ln 11-15 and Page 19, Ln 2-9).*

3) Further experiment needed on abiotic component (MBC or community structure) and will add values on the obtained data and comprehensive understanding on processes. This would certainly help us better understand the role of wetting and drying cycles on GHG (CO2, in this work).

*Authors' response: The authors agree that microbiological approaches would enable us to answer more questions regarding the microbes and metabolic pathways mineralizing carbon along our transects. However, this aspect is outside the scope of this manuscript, which is focused on the influence of mineral composition and redox state on carbon accumulation and emissions. Future efforts at the site will surely look into furthering our understanding of seasonal shifts in microbial community composition and function within these ecosystems.*

[revised manuscript text omitted]

fig01

[Figure]

fig01

[Figure]

fig02

[Figure]

fig03

(a)

[Figure]

(b)

[Figure]

fig04

[Figure]

fig05

[Figure]

fig06